# Core Context Aware Transformers for Long Context Language Modeling

**Yaofo Chen** [* 1]  **Zeng You** [* 1 2]  **Shuhai Zhang** [* 1 3]  **Haokun Li** [1 2]  **Yirui Li** [1]  **Yaowei Wang** [2 4]  **Mingkui Tan** [1 3 5]

## Abstract

Transformer-based Large Language Models (LLMs) have exhibited remarkable success in extensive tasks primarily attributed to the self-attention mechanism, which requires a token to consider all preceding tokens as its context to compute attention. However, when the context length $L$ becomes very large (*e.g.*, 128K), the amount of potentially redundant information in the context tends to increase. The redundant context not only hampers the modeling representation performance but also incurs unnecessary computational and storage overhead. In this paper, we propose a plug-and-play Core Context Aware (CCA) Attention for efficient long-context modeling, comprising two complementary modules: 1) *Globality-aware pooling module* groups input tokens and dynamically compresses each group into one *core token* based on their significance. In this way, our method automatically focuses and strengthens core context while diminishing redundancy during the learning process, leading to effective long-term dependency modeling. 2) *Locality-preserving module* incorporates neighboring tokens to preserve local context for detailed representation. Notably, our CCA-Attention is able to replace the self-attention module in existing LLMs with minimal fine-tuning cost. Extensive experimental results show the superiority of our method in both long-context modeling and computational efficiency over state-of-the-art methods.

---

[*]Equal contribution  [1]South China University of Technology [2]Peng Cheng Laboratory [3]Pazhou Laboratory [4]Harbin Institute of Technology [5]Key Laboratory of Big Data and Intelligent Robot, Ministry of Education. Correspondence to: Mingkui Tan <mingkuitan@scut.edu.cn>, Yaowei Wang <wangyw@pcl.ac.cn>.

*Proceedings of the 42$^{nd}$ International Conference on Machine Learning*, Vancouver, Canada. PMLR 267, 2025. Copyright 2025 by the author(s).

## 1. Introduction

Large language models (LLMs) (Brown et al., 2020; OpenAI, 2023; Touvron et al., 2023a; Liu et al., 2024a) have demonstrated exceptional proficiency across various applications by effectively modeling extended contexts, particularly in tasks involving natural language understanding and generation (Ouyang et al., 2022; Chang et al., 2024). The remarkable success of Transformer-based LLMs is predominantly credited to the self-attention mechanism (Vaswani et al., 2017), which requires each token to incorporate all preceding tokens as its context for attention calculation. In this mechanism, the context of a token refers to the sequence of tokens that precede it. By leveraging self-attention, LLMs are able to capture long-range dependencies and generate coherent and contextually relevant outputs.

The ability to process longer contexts has been a key factor in improving the performance of LLMs across a wide range of tasks, particularly beneficial for tasks requiring document-level understanding, such as summarization of extended texts, and multi-turn dialogue (Kitaev et al., 2019; Wei et al., 2022; Jiang et al., 2024). More importantly, recent advancements in LLMs, such as OpenAI-o1 (OpenAI, 2023) and DeepSeek-R1 (Guo et al., 2025), have demonstrated that extended contexts significantly enhance reasoning capabilities, enabling models to solve intricate problems that require multi-step inference and contextual understanding. This trend underscores the importance of extending the context length in LLMs to enhance modeling capabilities.

However, as the context length $L$ scales to extremely large magnitudes (*e.g.*, 128K), it becomes impractical for a token to maintain significant semantic connections with all tokens within such an extensive context. From this perspective, it is natural to consider that not all parts of the context contribute equally to a token's representation. Instead, the context can be viewed as comprising two primary aspects: core context, which captures essential semantic connections, and redundant context, which contains less critical or repetitive information. The redundant context may hamper LLMs from capturing dependencies among crucial tokens, degrading representation performance. In self-attention, this redundancy manifests as a highly sparse distribution of attention scores, with a substantial proportion disproportionately assigned to a limited number of tokens.

Such sparsity in attention score distribution has been observed across different attention heads in most layers of LLMs, as shown in recent studies (Beltagy et al., 2020; Zaheer et al., 2020; Xiao et al., 2024b). This sparsity in attention scores introduces unnecessary computational and storage overhead, especially for extremely long contexts.

To address the above issues, numerous studies have been advanced to eliminate the redundancy and enhance attention computational efficiency. StreamingLLM (Xiao et al., 2024b) and LM-Infinite (Han et al., 2023) simply maintain the attention over only the initial and last several tokens, ignoring the attention connection among remaining tokens. Besides, MInference (Jiang et al., 2024) introduces an efficient mixed attention mechanism comprising A-shape, vertical-slash, and block-sparse attentions, with the mixed attention patterns determined offline based on some samples. These methodologies typically involve computing only a portion of the attention to approximate full attention, thus compromising the connections among different tokens. In question-answering tasks, the crucial information can be located across any position in the input tokens. Consequently, it is crucial for the model to have the capability to leverage information from any position within the input text (Liu et al., 2024b). In this sense, these methods with fixed sparsity patterns may lead to incomplete comprehension of the long context. Therefore, how to ensure the information exchange among tokens in the attention while reducing the context redundancy is still an open question.

In this paper, we propose an efficient **Core Context Aware (CCA) Attention** mechanism, which is designed to efficiently capture both global and local dependencies within long contexts. Specifically, our CCA-Attention consists of two complementary components: 1) *Globality-aware pooling module* first partitions the input tokens into groups and derives *core tokens* by compressing the input tokens in each group based on their significance. We perform attention on these core tokens instead of original input tokens to efficiently extract long-term contextual information. These number-reduced core tokens are more compact representations than the original ones, which enables our attention method to automatically focus on the core context. In this way, our method is able to eliminate the context redundancy and reduce unnecessary computational overhead. However, the globality-aware pooling module is only able to capture long-range and coarse-grained information. 2) To address this issue, we propose a *Locality-preserving module* that captures the local and fine-grained context by focusing on neighboring tokens, serving as a complement for the globality-aware pooling module. By fusing the insights from both these two modules, our method not only excels in long context modeling but also achieves this with a significant reduction in computational costs and storage demands. Our contributions are as follows:

- We propose a plug-and-play Core Context Aware Attention for efficient long-context modeling. Our CCA-Attention reduces computational complexity to linear complexity by taking a set of core tokens as efficient proxies for attention. Unlike traditional efficient attention methods that require extensive retraining, our CCA-Attention can be easily integrated into pretrained LLMs with minimal fine-tuning effort.

- We develop a dynamic globality-aware pooling module that adaptively derives core tokens based on their importance. By compressing input tokens into core tokens, our method captures essential information more effectively than static or random token selection approaches. Our strategy focuses on the most relevant global context, leading to more accurate and effective long-term dependency modeling.

- We achieve significant improvements compared with other baseline methods in both long-context modeling performance and computational efficiency. Our experimental results show that CCA-Attention not only outperforms existing efficient attention mechanisms in long-context modeling but also achieves a $7.9\times$ speedup compared to full self-attention when processing 128K token contexts, demonstrating substantial efficiency gains with compatible accuracy.

## 2. Related Work

**Efficient Attention**. Self-attention is a fundamental module in Transformer-based Large Language Models (LLMs) (Brown et al., 2020; OpenAI, 2023; Touvron et al., 2023a). It captures the global relationship between each token throughout the input sequence. However, the computational complexity of self-attention increases quadratically with the sequence length, thereby limiting the application of LLMs to long documents. Various works have sought to mitigate this complexity through approaches such as sparse attention (Beltagy et al., 2020; Zaheer et al., 2020; Ding et al., 2023) and linear attention approximations (Choromanski et al., 2020; Katharopoulos et al., 2020; Sun et al., 2023). Specifically, Longformer (Beltagy et al., 2020) and BigBird (Zaheer et al., 2020) employ sparse attention mechanisms to handle long sequences by utilizing strided attention patterns, where attention is only paid at fixed intervals. Linear Transformer (Katharopoulos et al., 2020) and RetNet (Sun et al., 2023) reformulate self-attention as a linear dot-product of kernel feature maps and leverages the associativity property of matrix products to achieve linear complexity.

Recently, LongLoRA (Chen et al., 2024) designs a shifted sparse attention mechanism that computes attention among grouped input tokens. To facilitate communication between groups, this approach shifts the group partition by half the

*Figure 1.* Illustration of core contexts and redundant contexts. We show attention scores of the last token relative to the other tokens in LLaMA2-7B (darker shadows indicate higher attention scores). The last token exhibits high attention scores towards core contexts. The remains are considered as redundant contexts, introducing unnecessary computational overhead for attention.

group size. StreamingLLM (Xiao et al., 2024b) and LM-Infinite (Han et al., 2023) prioritize attention on the initial and final tokens, effectively disregarding the intermediate tokens. InfLLM (Xiao et al., 2024a) employs a sliding window attention mechanism and a block-level context memory to selectively attend to relevant context information, avoiding noise and reducing computational costs. MInference (Jiang et al., 2024) accelerates long-context LLM inference by applying three distinct sparse attention patterns with optimized GPU kernels. However, these methods can not ensure that each token has access to all preceding tokens, leading to inferior performance in tasks requiring comprehensive long-context understanding. Instead, we propose a globality-aware pooling module that each token can communicate with previous tokens via number-reduced core tokens.

**Long-context Large Language Models (LLMs)**. LLMs are often pretrained with a relatively small and predefined context length due to computational cost constraints, such as 4K for LLaMA2 (Touvron et al., 2023b). This limitation restricts the applicability of LLMs to tasks with long documents. Recently, several attempts have been made to extend the context length of LLMs through continuous training. Position Interpolation (Chen et al., 2023) addresses this by linearly down-scaling the input position indices to fit within the original context window size, thereby extending the context length of RoPE-based LLMs. Furthermore, YaRN (Peng et al., 2024) enhances performance by combining interpolation techniques with dynamic scaling. Beyond modifications to position embeddings, other efforts focus on designing more efficient attention mechanisms (Chen et al., 2024; Dao et al., 2022; Dao, 2024) for context window extension. Our method is orthogonal to position embedding methods. During inference, our approach accelerates the forward propagation process, which cannot be achieved through position embedding modifications alone. Some context compression works attempt to achieve long context modeling by either compressing features via auxiliary networks (Rae et al., 2020) or compressing context with extra specific tokens (Mu et al., 2023; Qin & Van Durme, 2023; Mohtashami & Jaggi, 2023; Zhang et al., 2024; Qin et al., 2024). In contrast, our method dynamically identifies and enhances task-relevant core-context while suppressing redundant information. Unlike sequence-length-oriented compression techniques, our core-context-aware mechanism optimizes redundancy directly within self-attention computation, enabling more effective long-context modeling.

## 3. Core Context Aware Attention

### 3.1. Motivation and Method Overview

Most existing attention-based large language models (LLMs), such as GPT (Brown et al., 2020; OpenAI, 2023) and LLaMA (Touvron et al., 2023a), employ the *next-token prediction* (Vaswani et al., 2017) paradigm to generate text. Given a sequence of tokens $\mathbf{X}=[\mathbf{x}_1; \mathbf{x}_2; \ldots; \mathbf{x}_L]$, where each token $\mathbf{x}_i \in \mathbb{R}^{1 \times d}$, the model $\theta$ predicts the next token $\mathbf{x}_t$ by conditioning on all preceding tokens as its context $C(\mathbf{x}_t)=[\mathbf{x}_{1:t-1}]$. Specifically, the model generates the next token with the highest probability as:

$$\mathbf{x}_t = \arg\max_{\mathbf{x}} P_\theta(\mathbf{x}|\mathbf{x}_1, \mathbf{x}_2, \ldots, \mathbf{x}_{t-1}). \quad (1)$$

However, as the context length $L$ grows, the context inevitably exhibits redundant information. This redundancy stems from the inherent nature of natural language: not all contextual information is equally important for the representation of the target token. These redundant context (*e.g.*, $C^1(\mathbf{x}_t)$ *w.r.t.* $\mathbf{x}_t$ in Figure 1) have weak semantic relevance to the token $\mathbf{x}_t$, while introducing significant computational overhead. In contrast, the core context (*e.g.*, $C^2(\mathbf{x}_t)$ *w.r.t.* $\mathbf{x}_t$ in Figure 1) refers to the contextual information that is highly relevant to the token $\mathbf{x}_t$. This information is crucial for the token's representation. Therefore, in long-context modeling, the model should prioritize the core context over redundant parts. To this end, most existing methods (Beltagy et al., 2020; Ding et al., 2023; Chen et al., 2024) employ sparse attention with predefined and fixed patterns. Unfortunately, they often overlook the importance of maintaining comprehensive information exchange among tokens, which may hinder the performance of long-context modeling tasks.

In this paper, we seek to reduce the context redundancy associated with full self-attention. To achieve this, we propose a Core Context Aware Attention (CCA-Attention), which employs globality-aware pooling and locality-preserving mod-

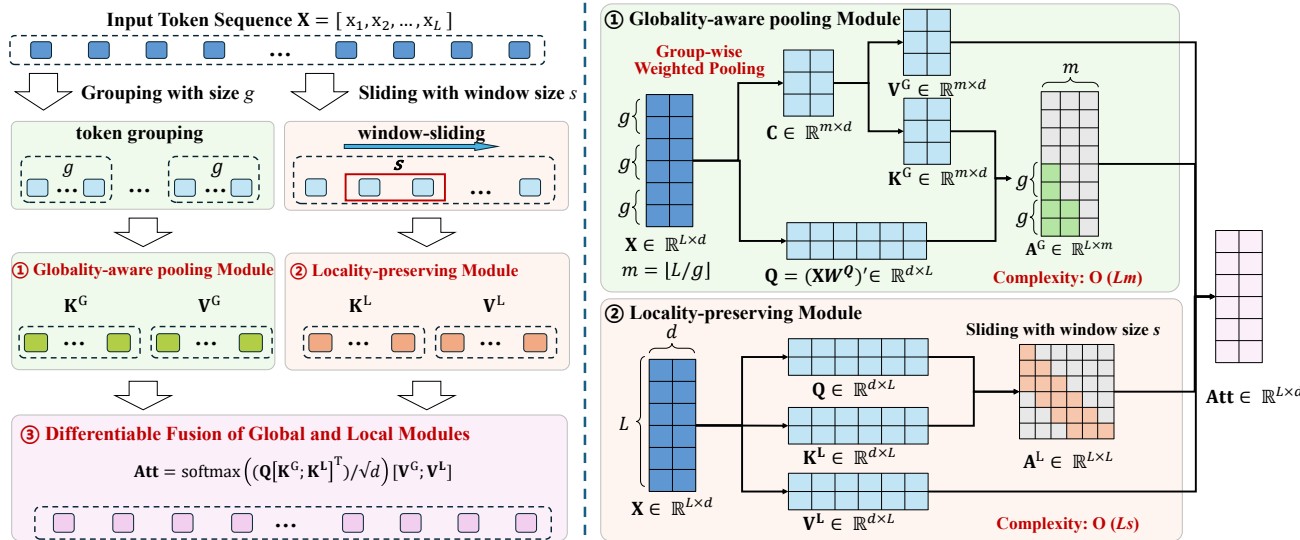

*Figure 2.* Illustration of CCA-Attention, which includes two components: 1) Globality-aware pooling module encapsulates the input tokens $\mathbf{X}$ into core tokens $\mathbf{C}$ according to the importance (Eqn. (2)). The core tokens $\mathbf{C}$ serve as representative proxies of $\mathbf{X}$ for attention, thereby reducing computational costs. 2) Locality-preserving module incorporates the local context from neighboring tokens, acting as supplement for the globality-aware pooling module. We produce the final output $\mathbf{Att}$ by fusing these two modules based on Eqn. (5).

ules to capture both global and local dependencies within a long context. As shown in Figure 2, the globality-aware pooling module operates by generating representative core tokens from segmented groups of the input sequence. It then computes attention using these reduced-number core tokens, thereby reducing the context redundancy and computational cost (see Section 3.2). However, the globality-aware pooling module mainly focuses on long-range and coarse-grained information and overlooks local context. To address this limitation, the locality-preserving module is responsible for capturing the local information of the neighborhood tokens to ensure comprehensive coverage (see Section 3.3). Furthermore, we devise a differentiable fusion strategy to combine the insights from global and local modules (see Section 3.4). This is crucial as it retains the comprehensive understanding ability of the full self-attention within our CCA-Attention. The pseudo-code for our proposed CCA-Attention is presented in Algorithm 1.

### 3.2. Globality-aware Pooling Module

The context redundancy aforementioned indicates that computational resources can be dynamically allocated to core contexts while reducing emphasis on the remaining ones. This could approximate the full self-attention with both reduced redundancy and computational overhead. Motivated by this, we propose a globality-aware pooling module that dynamically identifies prominent tokens and encapsulates them into a smaller set of core tokens for attention.

Given an input sequence of tokens $\mathbf{X}=[\mathbf{x}_1;\mathbf{x}_2;\ldots;\mathbf{x}_L]$, we

segment the input sequence $\mathbf{X}$, each group containing $g$ tokens, in total $m=\lfloor L/g \rfloor$ groups. For simplicity, we denote the $i$-th group by $\mathbf{X}_{\mathcal{I}_i}^{\mathrm{G}} \in \mathbb{R}^{g \times d}$, where $\mathcal{I}_i = \{(i-1)g + 1, (i-1)g+2, \ldots, ig\}$ with $\mathbf{x}_{ig}$ denoting the last token in the $i$-th group. To identify prominent tokens in the $i$-th group, we devise a group-wise weighted pooling strategy that employs the last token $\mathbf{x}_{ig}$ to evaluate the importance. This is inspired by attention map visualizations (Section C.4), which show that important tokens consistently receive high attention scores from subsequent tokens, indicating their significant influence regardless of position within the group. Formally, we derive one core token $\mathbf{c}_i$ from each group by

$$\mathbf{c}_i = \mathrm{softmax}\left(\frac{\mathbf{Q}_{ig}\mathbf{K}_{\mathcal{I}_i}^{'\top}}{\sqrt{d}}\right)\mathbf{X}_{\mathcal{I}_i}^{\mathrm{G}} \in \mathbb{R}^{1 \times d}, i = 1, \ldots, m,$$

$$(2)$$

where $\mathbf{Q}_{ig}$ is the query vector for the last token in the $i$-th group of $\mathbf{Q}_{\mathcal{I}_i}=\mathbf{X}_{\mathcal{I}_i}^{\mathrm{G}}\mathbf{W}^Q$ and $\mathbf{K}_{\mathcal{I}_i}^{'}=\mathbf{X}_{\mathcal{I}_i}^{\mathrm{G}}\mathbf{W}^K$, $\mathbf{W}^Q$ and $\mathbf{W}^K$ are learnable parameters. In this way, the core token $\mathbf{c}_i$ encapsulates crucial information of the corresponding group. With $m$ groups in the input sequence $\mathbf{X}$, we derive $m$ core tokens in total, *i.e.*, $\mathbf{C}=[\mathbf{c}_1;\mathbf{c}_2;\ldots;\mathbf{c}_m]$.

To reduce the redundancy, we use the sequence of core tokens $\mathbf{C}=[\mathbf{c}_1;\mathbf{c}_2;\ldots;\mathbf{c}_m]$ instead of the original tokens $\mathbf{X}$ for attention computation. This substitution reduces the dimensionality from $\mathbf{X}\in\mathbb{R}^{L \times d}$ to $\mathbf{C}\in\mathbb{R}^{m \times d}$, thereby reducing the computational and storage complexity. For each query $\mathbf{Q}_i$, tokens that are distant from it typically exhibit lower relevance and are more likely to be redundant. Formally, we adopt core tokens to calculate the key and value matrices

**Algorithm 1** The pipeline of Core Context Aware Attention.

---

**Require:** Input tokens $\mathbf{X}=[\mathbf{x}_1;\mathbf{x}_2;\ldots;\mathbf{x}_L]$, parameters $\mathbf{W}^Q$, $\mathbf{W}^K$, $\mathbf{W}^V$, the group size $g$ and local window size $s$.
1: Calculate the query $\mathbf{Q}=\mathbf{XW}^Q$, #groups $m=\lfloor L/g \rfloor$
2: **for** $i$ in $\{1,2,\ldots,m\}$ **do**
3:    $\mathbf{X}^{\mathrm{G}}_{\mathcal{I}_i}=[\mathbf{x}_{(i-1)g+1};\mathbf{x}_{(i-1)g+2};\ldots;\mathbf{x}_{ig}]$
4:    $\mathbf{c}_i=\mathrm{softmax}\left(\frac{\mathbf{Q}_{ig}\mathbf{K}^{'\top}_{\mathcal{I}_i}}{\sqrt{d}}\right)\mathbf{X}^{\mathrm{G}}_{\mathcal{I}_i}$, where $\mathbf{K}^{'}_{\mathcal{I}_i}=\mathbf{X}^{\mathrm{G}}_{\mathcal{I}_i}\mathbf{W}^K$
5: **end for**
6: Let $\mathbf{C}=[\mathbf{c}_1;\mathbf{c}_2;\ldots;\mathbf{c}_m]$
7: $\mathbf{K}^{\mathrm{G}}=\mathbf{CW}^K$, $\mathbf{V}^{\mathrm{G}}=\mathbf{CW}^V$ // *Globality-aware Pooling Module*
8: $\mathbf{K}^{\mathrm{L}}=\mathbf{XW}^K$, $\mathbf{V}^{\mathrm{L}}=\mathbf{XW}^V$ // *Locality-preserving Module*
9: **for** $i$ in $\{1,2,\ldots,L\}$ **do**
10:    $\widetilde{\mathbf{K}}^{\mathrm{G}}_{\mathcal{T}_i}=\mathbf{K}^{\mathrm{G}}_{1:j}$, $\widetilde{\mathbf{V}}^{\mathrm{G}}_{\mathcal{T}_i}=\mathbf{V}^{\mathrm{G}}_{1:j}$, $j=\max(0,\lfloor(i-s)/g\rfloor)$
11:    $\widetilde{\mathbf{K}}^{\mathrm{L}}_{\mathcal{U}_i}=\mathbf{K}^{\mathrm{L}}_{k:i}$, $\widetilde{\mathbf{V}}^{\mathrm{L}}_{\mathcal{U}_i}=\mathbf{V}^{\mathrm{L}}_{k:i}$, $k=\max(1,i-s-((i-s)\bmod g))$
12:    $\mathbf{Att}_i=\mathrm{softmax}(\mathbf{Q}_i[\widetilde{\mathbf{K}}^{\mathrm{G}}_{\mathcal{T}_i};\widetilde{\mathbf{K}}^{\mathrm{L}}_{\mathcal{U}_i}]^\top)/\sqrt{d})[\widetilde{\mathbf{V}}^{\mathrm{G}}_{\mathcal{T}_i};\widetilde{\mathbf{V}}^{\mathrm{L}}_{\mathcal{U}_i}]$
13: **end for**
14: **Return:** Representations of tokens $\mathbf{Att}=[\mathbf{Att}_1;\ldots;\mathbf{Att}_L]$

---

$\mathbf{K}^{\mathrm{G}}$, $\mathbf{V}^{\mathrm{G}}$ for these tokens as follows

$$\widetilde{\mathbf{K}}^{\mathrm{G}}_{\mathcal{T}_i}=[\mathbf{K}^{\mathrm{G}}_1;\cdots;\mathbf{K}^{\mathrm{G}}_j], \widetilde{\mathbf{V}}^{\mathrm{G}}_{\mathcal{T}_i}=[\mathbf{V}^{\mathrm{G}}_1;\cdots;\mathbf{V}^{\mathrm{G}}_j],$$
$$\mathbf{K}^{\mathrm{G}}=\mathbf{CW}^K, \mathbf{V}^{\mathrm{G}}=\mathbf{CW}^V, \qquad (3)$$

where $\mathbf{W}^K$ and $\mathbf{W}^V$ is learnable parameters. In contrast, tokens in close proximity to the query $\mathbf{Q}_i$ are likely to be more relevant. We retain $s$ nearest tokens for a fine-grained attention computation (discussed in detail in Section 3.3). Thus, the index $j$ in Eqn. (3) can be calculated as $j=\max(0,\lfloor(i-s)/g\rfloor)$. When the context is short (*i.e.*, $i<(g+s)$), the key and value $\mathbf{K}^{\mathrm{G}}$, $\mathbf{V}^{\mathrm{G}}$ would be excluded from attention calculation since the redundancy in the context is negligible. During inference, as tokens are generated sequentially, we derive a new core token via Eqn. (2) once the number of generated tokens reaches $g$. Different from the full self-attention, we cache $\widetilde{\mathbf{K}}^{\mathrm{G}}$ and $\widetilde{\mathbf{V}}^{\mathrm{G}}$ for inference.

### 3.3. Locality-preserving Module

As mentioned above, the globality-aware pooling module effectively captures long-range dependencies by compressing input tokens into core tokens. It focuses on coarse-grained global information, potentially overlooking fine-grained local context. However, recent studies (Manakul & Gales, 2021; Yang et al., 2021) demonstrate that local context plays a critical role in many language modeling tasks. To address this, we introduce a locality-preserving module that complements the globality-aware pooling module by focusing on neighboring tokens to capture detailed local dependencies.

To be specific, the locality-preserving module ensures that each query $\mathbf{Q}_i$ attends to the preceding at least $s$ tokens to capture local dependencies. During the generation process, it is challenging to maintain the number of tokens as a multiple of the group size $g$. To address this, we set the local

window size to $s+((i-s)\bmod g)$. This strategy ensures that each key token participates in the attention computation with the query $\mathbf{Q}_i$. Consequently, the key and value matrices for a specific query $\mathbf{Q}_i$ in the locality-preserving module are defined as follows:

$$\widetilde{\mathbf{K}}^{\mathrm{L}}_{\mathcal{U}_i}=[\mathbf{K}^{\mathrm{L}}_k;\cdots;\mathbf{K}^{\mathrm{L}}_i], \widetilde{\mathbf{V}}^{\mathrm{L}}_{\mathcal{U}_i}=[\mathbf{V}^{\mathrm{L}}_k;\cdots;\mathbf{V}^{\mathrm{L}}_i],$$
$$\mathbf{K}^{\mathrm{L}}=\mathbf{XW}^K, \mathbf{V}^{\mathrm{L}}=\mathbf{XW}^V \qquad (4)$$

where $k=\max(1,i-s-((i-s)\bmod g))$. Note that the locality-preserving module shares the projection parameters $\mathbf{W}^Q$, $\mathbf{W}^K$, and $\mathbf{W}^V$ with the globality-aware pooling module, thereby incurring no additional projection parameters.

### 3.4. Differentiable Fusion of Global and Local Modules

Both globality-aware pooling and locality-preserving modules involve only a portion of tokens in the attention computation, leading to a limited comprehensive understanding of the context. To address this limitation, we seek to combine the involved tokens of these two attentions to integrate the insights they provide. Specifically, we concatenate the key and value matrices from both attentions, *i.e.*, $[\widetilde{\mathbf{K}}^{\mathrm{G}}_{\mathcal{T}_i};\widetilde{\mathbf{K}}^{\mathrm{L}}_{\mathcal{U}_i}]$ and $[\widetilde{\mathbf{V}}^{\mathrm{G}}_{\mathcal{T}_i};\widetilde{\mathbf{V}}^{\mathrm{L}}_{\mathcal{U}_i}]$, to leverage the combined information. Formally, the proposed CCA-Attention is computed as follows:

$$\mathbf{Att}_i=\mathrm{softmax}\left(\frac{\mathbf{Q}_i[\widetilde{\mathbf{K}}^{\mathrm{G}}_{\mathcal{T}_i};\widetilde{\mathbf{K}}^{\mathrm{L}}_{\mathcal{U}_i}]^\top)}{\sqrt{d}}\right)[\widetilde{\mathbf{V}}^{\mathrm{G}}_{\mathcal{T}_i};\widetilde{\mathbf{V}}^{\mathrm{L}}_{\mathcal{U}_i}]. \qquad (5)$$

We represent the final output of our CCA-Attention as $\mathbf{Att}=[\mathbf{Att}_1;\mathbf{Att}_2;\ldots;\mathbf{Att}_L]$. In practice, we implement our attention mechanism with Triton (Tillet et al., 2019) to accelerate both training and inference processes. This implementation enables parallel computation of each $\mathbf{Att}_i$ with high computational efficiency. After integrating the global and local attention, we can formalize $\mathbf{Att}$ in Eqn. (5) element-wise into the structure of full attention, as detailed in Proposition 1 in the supplementary material A. The Proposition 1 shows that each token accesses all preceding tokens, ensuring full information exchange among tokens and thus enhancing capturing long-range dependencies.

More importantly, our CCA-Attention demonstrates dynamic flexibility through adjustable group size $g$ and local window size $s$ during inference. This architectural flexibility allows the generation of multiple model variants tailored to varying user traffic, offering a substantial advantage over the full self-attention mechanisms in real-world deployment scenarios (see results and analysis in Section 4.3).

### 3.5. Training Strategies of CCA Transformers

Our proposed CCA-Attention is fully compatible with existing attention-based LLMs, such as the LLaMA series models (Touvron et al., 2023b; AI, 2024), serving as a plug-and-play module that can replace the full self-attention.

CCA-Attention maintains alignment with full self-attention in terms of input, output, and parameter dimensions. This ensures that only a minimal training cost is able to preserve long-context modeling capabilities while reducing computational costs. In contrast, existing linear attention approaches (Katharopoulos et al., 2020; Sun et al., 2023) introduce kernel functions for attention and necessitate training from scratch, making them less practical for real-world applications due to their inability to leverage the extensive knowledge embedded in pretrained LLMs.

We replace the self-attention module in existing attention-based LLMs with CCA-Attention, enabling compatibility with three different training strategies: 1) Training from Scratch: This strategy involves training CCA-Attention from scratch on large-scale corpora. While this approach may yield superior performance by fully adapting the model to the proposed attention mechanism, it is computationally intensive and requires significant resources. 2) Full Finetuning: A more efficient alternative is to finetune all parameters of the model based on the parameters of existing pretrained LLMs. This strategy leverages the pre-trained knowledge while allowing the model to adapt fully to our attention mechanism. 3) Partial Finetuning: For scenarios where efficiency is critical, we can finetune only the learnable parameters of CCA-Attention, *i.e.*, $\mathbf{W}^Q$, $\mathbf{W}^K$, and $\mathbf{W}^V$. This strategy requires only a modest finetuning effort on a small number of corpora, making it computationally efficient for maintaining long-context modeling capabilities. We provide empirical evidence to guide the selection of the most appropriate finetuning strategy in Table 10.

### 3.6. Computational and Storage Complexity Analysis

Compared with the full self-attention, our CCA-Attention offers significant benefits in terms of computational complexity and key-value cache storage, as analyzed below.

**Acceleration via Reduced Computational Complexity**. Our CCA-Attention exhibits varying computational complexities depending on the type of task. For tasks with fixed-length sequences (such as multi-choice question answering), our CCA-Attention exhibits a linear computational complexity of $O(Lm + Ls)$, marking a significant enhancement over the full self-attention with a complexity of $O(L^2)$. Here, we define the number of group $m$ as a constant. For the globality-aware pooling module, the query and key matrices encompass $L$ and $m$ tokens, respectively, resulting in a computational complexity of $\mathcal{O}(Lm)$. Regarding the locality-preserving module, each token only attends preceding $s + g$ tokens at most. With $L$ tokens in total, the upper bound of complexity amounts to $O(L(s + g))$.

For tasks with variable-length sequences (such as open-ended question answering), models generate subsequent tokens in an autoregressive manner. In this case, we set the group size $g$ as a constant, ensuring that our CCA-Attention is able to leverage key-value caching during autoregressive token generation. Once one group has certain $g$ tokens, the corresponding core token is also determined and cached. Thus, our CCA-Attention achieves a computational complexity of $O(L^2/g + Ls)$. The complexity analysis follows a similar pattern to the tasks with fixed-length sequences.

**Acceleration through Reduced Key-Value (KV) Cache**. In attention-based LLMs, the KV cache leverages the autoregressive nature to store and reuse key-value pairs, thereby significantly boosting the efficiency. The size of the KV cache scales linearly with the length of the input sequence, consuming a major part of the memory footprint during inference. The expanded KV cache would consume considerable memory and significant memory IO resources. Compared with full attention's complexity of $O(L)$, our CCA-Attention has a storage complexity of $O(L/g + s)$. For the globality-aware pooling module, we only retain the key and value matrices for core tokens, rather than for all original tokens. This reduces the memory requirement to $O(L/g)$. Besides, the locality-preserving module only maintains the key and value matrices for the preceding $s$ tokens. The storage complexity for this component is $O(s)$.

## 4. Experiments

### 4.1. Experimental Setup

We apply our CCA-Attention and compared efficient attention methods to existing pretrained LLMs. We report the performance in long context modeling and computational efficiency. We put more implementation details and ablation studies of our method in the supplementary materials.

**Dataset & Evaluation Metrics**. We quantitatively evaluate our models and compare them with other considered models on the following benchmark and metric: 1) LongBench (Bai et al., 2023) is a pioneering benchmark for the bilingual, multi-task, and comprehensive assessment of large language models' long context understanding capabilities. It covers multiple languages like Chinese and English, consisting of 6 major categories and 21 tasks involving various application areas. 2) Exact Match Score (EM Score) (Liu et al., 2024b) is a metric for measuring the model's ability to find the key information within a long context in a multi-document question-answering task. In this task, each test sample comprises a certain number of documents to reach the specified context length, followed by a question about the key information inserted in context.

**Implementation Details**[1]. We apply our proposed CCA-Attention to LLaMA2-7B-32K, LLaMA2-7B-80K (Fu et al.,

---

[1]The source code for this project is publicly available at https://github.com/chenyaofo/CCA-Attention.

*Table 1.* Comparisons of different models on LongBench-E (Bai et al., 2023). The length of 95% of the test samples in LongBench-E is less than 31K. "FTL" denotes the latency to generate the first token in the pre-filling stage. "Mem." denotes the memory footprint. "S. QA" means single document QA, while "M. QA" denotes multi-document QA. We report the latency and memory footprint of LLaMA2-7B-32K and LLaMA2-7B-80K within contexts of 32K and 64K on A800 GPUs, respectively.

| Methods | S. QA | M. QA | Sum. | FS. Learning | Synthetic | Code | Avg. | FTL (s) | Mem. (GB) |
|---|---|---|---|---|---|---|---|---|---|
| *LLaMA2-7B-32K (Vanilla Self-Attention)* | 2.75 | 1.85 | **12.43** | **66.28** | 0.34 | 48.99 | **22.11** | 9.15 | 35.58 |
| StreamingLLM (Xiao et al., 2024b) | **4.75** | 2.94 | 2.97 | 48.20 | 0.66 | 30.16 | 14.95 | 5.75 (1.6×) | 22.94 (35%↓) |
| LM-Infinite (Han et al., 2023) | 2.04 | 2.33 | 1.98 | 57.45 | 0.3 | 48.46 | 18.76 | 4.72 (1.9×) | 26.35 (26%↓) |
| MInference (Jiang et al., 2024) | 3.68 | 3.05 | 10.97 | 66.26 | 0.61 | 42.30 | 21.14 | 4.20 (2.2×) | 33.52 (6%↓) |
| CCA-LLM (Ours) | 3.63 | **3.98** | 7.79 | 61.79 | **2.64** | **51.36** | 21.86 | **2.59 (3.5×)** | **19.12 (46%↓)** |
| *LLaMA2-7B-80K (Vanilla Self-Attention)* | 3.22 | 2.71 | 3.90 | **64.98** | 0.56 | **59.16** | 22.42 | 32.43 | 60.03 |
| StreamingLLM (Xiao et al., 2024b) | 2.07 | 2.32 | 0.37 | 45.03 | **2.67** | 37.17 | 14.94 | 9.04 (3.6×) | 37.45 (37%↓) |
| LM-Infinite (Han et al., 2023) | 2.54 | 1.53 | 2.22 | 61.29 | 1.08 | 58.54 | 21.20 | 8.27 (3.9×) | 41.54 (31%↓) |
| MInference (Jiang et al., 2024) | 2.44 | 3.49 | 4.41 | 64.26 | 0.28 | 57.60 | 22.08 | 8.14 (4.0×) | 54.09 (10%↓) |
| CCA-LLM (Ours) | **5.62** | **4.34** | **8.99** | 59.60 | 0.48 | 54.40 | 22.24 | **6.42 (5.7×)** | **33.86 (44%↓)** |

*Table 2.* Comparisons of different methods using latest models on LongBench-E (Bai et al., 2023). We report the latency and memory footprint of LLaMA3.1-8B-Instruct-128K (short for "LLaMA3.1-8B-128K" in the table) and Qwen2.5-7B-128K within contexts of 32K on A800 GPUs.

| Methods | S. QA | M. QA | Sum. | FS. Learning | Synthetic | Code | Avg. | FTL (s) | Mem. (GB) |
|---|---|---|---|---|---|---|---|---|---|
| *LLaMA3.1-8B-128K (Vanilla Self-Attention)* | 16.71 | 10.75 | 20.32 | **68.75** | **48.93** | 62.10 | **37.93** | 9.55 | 40.38 |
| MInference (Jiang et al., 2024) | 16.33 | 10.71 | **20.44** | 68.41 | 48.06 | **62.50** | 37.74 | 4.93 (1.9×) | 35.95 (11%↓) |
| CCA-LLM (Ours) | **17.90** | **16.41** | 19.63 | 67.20 | 43.76 | 61.98 | 37.81 | **3.08 (3.1×)** | **20.63 (49%↓)** |
| *Qwen2.5-7B-128K (Vanilla Self-Attention)* | 16.67 | **18.18** | **18.70** | 66.81 | 45.34 | **64.56** | **38.38** | 10.58 | 35.11 |
| MInference (Jiang et al., 2024) | 16.20 | 17.21 | 18.59 | **67.10** | 38.28 | 62.95 | 36.72 | 4.86 (2.2×) | 32.40 (8%↓) |
| CCA-LLM (Ours) | **16.91** | 17.07 | 18.60 | 66.89 | **45.50** | 63.52 | 38.08 | **2.74 (3.9×)** | **19.31 (45%↓)** |

2024), LLaMA3.1-8B-128K (AI, 2024) and Qwen2.5-7B-128K (Yang et al., 2024) models. We replace the full self-attention in the above LLMs with our proposed CCA-Attention. In the continuous finetuning, we adopt the SlimPajama (Cerebras, 2024) dataset, an open-source replication of the LLaMA pretraining data mixture. The number of groups in globality-aware Attention is shared across different model sizes. We finetune the full model on A800 GPUs using a micro-batch size of 1 and a gradient accumulation of 8, with a total of 1000 training steps. This training configuration is applicable to all model sizes and context lengths. To scale the models to long contexts, we modified the "base frequency" in RoPE from 10,000 to 500,000, following (Cerebras, 2024; Xiong et al., 2024). See Section B.2 for more implementation details.

**Compared Methods**. We conduct comprehensive comparisons between our proposed CCA-Attention and several state-of-the-art methods, including LLaMA-2 with vanilla attention, StreamingLLM (Xiao et al., 2024b), LM-infinite (Han et al., 2023), and MInference (Jiang et al., 2024), across the LongBench (Bai et al., 2023) and RULER (Hsieh et al., 2024) benchmarks. Our experiments are based on the LLaMA-2 7B model fine-tuned on sequences of length 32K and 80K (Fu et al., 2024). For StreamingLLM (Xiao et al., 2024b), we use the official implementation, adjusting the attention sink to 4 and setting the attention context size to 2000. Similarly, for LM-infinite (Han et al., 2023), we follow the official code, con-

figuring the local branch size to 1024 and the global branch size to 16. In the case of MInference (Jiang et al., 2024), we also employ the official code implementations, configured with the official settings.

### 4.2. Comparisons on Long Context Modeling

**Comparisons on Longbench-E**. We conduct experiments on Longbench-E (Bai et al., 2023) using our CCA-Attention and baseline methods, including StreamingLLM (Xiao et al., 2024b), LM-Infinite (Han et al., 2023), and MInference (Jiang et al., 2024). As shown in Table 1, our CCA-LLM attains the highest average score on Longbench-E, outperforming other efficient attention methods. For LLaMA-7B-32K, the average score of our CCA-LLM is higher than that of LM-Infinite (21.86 *vs.* 18.76) and MInference (21.86 *vs.* 21.14). For the LLaMA-7B-80K model, our method consistently shows superior performance compared to alternative approaches. For instance, our CCA-LLM yields a higher EM score than LM-Infinite (22.24 *vs.* 21.20) and MInference (22.24 *vs.* 22.08). This performance advantage primarily stems from our global-aware pooling module, which effectively reduces context redundancy in long input sequences. Consequently, our CCA-LLM model demonstrates enhanced capability in identifying and focusing on core contextual elements, thereby facilitating more accurate extraction of crucial information required for question-answering tasks within the Longbench-E benchmark. Notably, our CCA-LLM achieves the lowest inference latency

*Table 3.* Comparisons of different models on multi-document EM score (Liu et al., 2024b) evaluated at lengths from 4K to 128K. "FTL" denotes the latency to generate the first token in the pre-filling stage. We report the latency within a context of 128K on two A800 GPUs.

| Methods | 4K | 8K | 16K | 32K | 64K | 128K | Avg. | FTL (s) |
|---|---|---|---|---|---|---|---|---|
| *LLaMA2-7B-80K (Vanilla Self-Attention)* | **39.4** | **37.8** | **37.6** | **36.2** | 34.6 | 30.3 | **36.0** | 124.85 |
| StreamingLLM (Xiao et al., 2024b) | 33.6 | 26.0 | 32.2 | 30.6 | 27.4 | 25.1 | 29.2 | 34.74 (3.6×) |
| LM-Infinite (Han et al., 2023) | 31.6 | 25.6 | 32.4 | 32.2 | 28.2 | 26.3 | 29.4 | 32.57 (3.8×) |
| MInference (Jiang et al., 2024) | 39.0 | 32.4 | 37.4 | 36.0 | 32.3 | 28.9 | 34.3 | 20.18 (6.2×) |
| CCA-LLM (Ours) | 39.3 | 33.2 | 35.4 | 31.4 | **35.3** | **32.0** | 34.4 | **15.89 (7.9×)** |

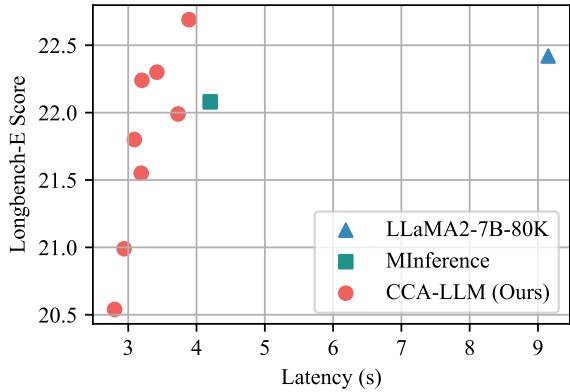

*Figure 3.* Illustration of inference flexibility by adjusting the group size $g$ and local window size $s$ to generate various CCA-LLM models with different latency and accuracies in the test time. This architectural flexibility allows for precise control over the trade-off between inference latency and accuracy, particularly beneficial for real-world applications with varying user traffic patterns.

and KV cache usage among all compared methods. ur CCA-LLM is able to reduce the KV cache usage while the best counterpart MInferencce only accerlerate the prefilling stage and does not reduce KV cache storage.

To further validate the effectiveness of our method across different model architectures, we conduct extensive experiments on more recent foundation models: LLaMA3.1-8B-Instruct-128K and Qwen2.5-7B-128K. As shown in Table 2. our CCA-Attention demonstrates superior performance over the state-of-the-art baseline MInference across three key metrics: computational efficiency, memory reduction, and model performance. The consistent improvements across different model architectures suggest that our approach is not limited to specific model designs.

**Comparisons on Long-document QA**. We evaluate the performance of our CCA-Attention against other methods, including StreamingLLM (Xiao et al., 2024b), LM-Infinite (Han et al., 2023), and MInference (Jiang et al., 2024), on LLaMA2-7B-80K models using the multi-document EM Score metric. As shown in Table 3, we conduct comparisons across a range of context lengths: 4K, 8K, 16K, 32K, 64K, and 128K. For the short context (*e.g.*, 4K), our CCA-LLM consistently achieves the highest EM score across all methods, showcasing significant capability

for short sequence modeling. For example, our CCA-LLM outperforms StreamLLM (39.3 *vs.* 33.6) and MInference (39.3 *vs.* 39.0) in terms of EM score. The reason is that our CCA-Attention captures context dependencies without discarding crucial tokens. In contrast, StreamingLLM (Xiao et al., 2024b) prioritizes attention on the initial and final tokens, effectively discarding intermediate tokens, which may contain essential information. Similarly, MInference (Jiang et al., 2024) employs predefined sparse attention patterns, selectively attending to tokens and potentially overlooking critical parts of the input sequence. Both approaches risk losing important contextual information, leading to suboptimal performance in tasks requiring comprehensive understanding. Our method, by preserving both local and global contexts, ensures that no critical information is overlooked, thereby achieving superior performance.

For the extremely long context (*e.g.*, 64K and 128K), Our CCA-LLM shows much better performance than vanilla self-attention in terms of EM score (35.3 *vs.* 34.6) and 7.9x inference speedup with a context length of 128K. The advantages of our method become more prominent as the length of the context increases, while the performance of vanilla self-attention may even decrease when the context length becomes very large. The reason is that in an extremely long context, non-core contexts (*i.e.*, the irrelevant context) will be compressed by the proposed weighted pooling. In this way, CCA-LLM alleviates the redundant context issue and improves the long-context modeling performance.

### 4.3. Demonstration of Inference Flexibility

In real-world applications, user traffic exhibits diurnal variations. During peak traffic periods, the full self-attention struggles with throughput limitations, necessitating additional servers to accommodate the increased demand. Instead, our CCA-LLM can dynamically adjust the group size $g$ and the local window size $s$ during inference to improve throughput. During off-peak traffic periods, our CCA-LLM is able to enhance model performance, albeit with a minor compromise in throughput. This strategy exhibits remarkable elasticity to address the variable user traffic challenges.

To verify this, we conduct experiments with different group sizes $g$ and local window sizes $s$ during inference. In Figure 3, our CCA-LLM demonstrates an improved Longbench-

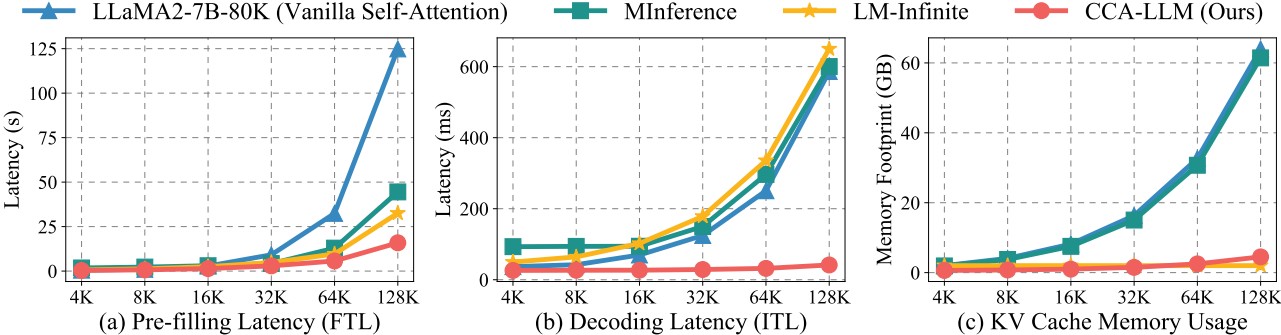

*Figure 4.* Comparisons with state-of-the-art methods in terms of both computational and storage overhead on LLaMA2-7B-80K. "FTL" (first token latency) is the time taken to generate the first token after receiving the input in the pre-filling stage. "ITL" (inter token latency) is the time delay between generating consecutive tokens (except for the first token) during the decoding stage.

E Score with a concomitant increase in computational overhead as $g$ decreases. With a reduction in the group size $g$ from 16 to 2 and the local window size $s$ from 4096 to 1024, the Longbench-E Score escalates from 20.54 to 22.69, while the latency rises from 2.80 seconds to 3.89 seconds. Despite training CCA-LLM only once, we obtain a spectrum of models, each with distinct performance and computational demands. This flexibility comes from two complementary modules: The globality-aware pooling module dynamically compress core tokens based on semantic relevance via intra-group attention, enabling adaptation to different group sizes. The locality-preserving module uses rotary embeddings to encode relative positions, achieving translation invariance and maintaining local context across scales.

### 4.4. Comparisons on Computational Efficiency

We compare our CCA-LLM with LLaMA2-7B-80K equipped with full self-attention, LM-Infinite (Han et al., 2023), and MInference (Jiang et al., 2024) in terms of inference latency and memory footprint during forward-propagation on a single NVIDIA A800 GPU. The efficiency performance was assessed across a range of input sequence lengths, *i.e.*, {4K, 8K, 16K, 32K, 64K, 128K}. In Figure 4, our CCA-Attention achieves a 7.9× inference speedup than LLaMA2-7B-80K with the full self-attention (15.89s *vs.* 124.85s in 128K context) MInference (15.89s *vs.* 44.50s in 128K context) in the pre-filling stage. Our CCA-Attention also exhibits a reduced KV cache memory usage than LLaMA2-7B-80K (4.5GB *vs.* 64GB in 128K context) and MInference (4.5GB *vs.* 64GB in 128K context). Note that Minference (Chen et al., 2024) only accelerates the pre-filling stage and adopts full self-attention in the decoding stage. Moreover, it does not reduce KV cache, leading to the same memory usage as the full self-attention. Conversely, our CCA-LLM is able to accelerate both pre-filling and decoding stages with reduced KV cache, which is more practical in real-world applications.

## 5. Conclusion

In this paper, we proposed a Core Context Aware Attention (CCA-Attention) for long-context language modeling with reduced computational overhead compared with vanilla self-attention. Our CCA-Attention includes two components: 1) globality-aware pooling module exploits the importance of input tokens to encapsulates core tokens and employs them for attention, capturing global coarse-grained information; 2) The locality-preserving module focuses on neighboring tokens to capture local fined-grained context, serving as a complement for the global module. Our proposed attention is able to replace the full self-attention in existing LLMs with a minimal finetuning efforts. Experimental results show the effectiveness of our CCA-Attention with promising performance and decreased computational cost.

## Acknowledgments

This work was partially supported by the Joint Funds of the National Natural Science Foundation of China (Grant No.U24A20327), Key-Area Research and Development Program Guangdong Province 2018B010107001, the Major Key Project of Peng Cheng Laboratory (PCL) PCL2023A08, Postdoctoral Fellowship Program of CPSF (GZC20251043), and TCL Science and Technology Innovation Fund, China.

## Impact Statement

Our work addresses the critical challenge of computational inefficiency in long-context modeling for large language models. By introducing a core context aware attention mechanism, we achieve a reduction in attention complexity to linear time while preserving essential semantic interactions. This enables LLMs to process contexts up to 128K tokens with a 7.9× speedup over full self-attention, without compromising accuracy. This promotes a more environmentally friendly and sustainable approach to AI development, aligning with the growing demand for more energy-efficient AI.

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

# Supplementary Materials

## Contents

# A. Theoretical Analysis on Reachability for CCA-Attention

In the self-attention, the calculation can be formulated as $\text{Attention}(\mathbf{Q}, \mathbf{K}, \mathbf{V}) = \text{softmax}\left(\mathbf{Q}\mathbf{K}^\top/\sqrt{d}\right)\mathbf{V}$, where $\mathbf{Q}=\mathbf{X}\mathbf{W}^Q$, $\mathbf{K}=\mathbf{X}\mathbf{W}^K$, and $\mathbf{V}=\mathbf{X}\mathbf{W}^V$, $\mathbf{W}^Q$, $\mathbf{W}^K$, $\mathbf{W}^V$ are learable parameters. For convenience in analyzing the attention mechanism, we denote the attention weight as $\mathbf{A} = \text{softmax}(\mathbf{Q}\mathbf{K}^\top/\sqrt{d})$, where the element in $\mathbf{A}$ is represented as $a_{ij}$. We give the definition of *reachability*, and show that our method is to satisfy the information among all tokens is reachability, and then give the concrete expression of attention output.

**Definition 1.** (**Reachability**) *We say the token $j$ is reachable from the token $i$ in the attention map if and only if the attention weight from the token $j$ to $i$ is positive, i.e., $a_{ij} > 0$.*

**Proposition 1.** *The attention score with causal masking in the CCA-Attention mechanism fully satisfies the reachability condition from the earlier tokens to the later tokens in the sequence at each transformer layer. Moreover, the final output representation $\mathbf{o} \in \mathbb{R}^d$ for $i$-th token in Eqn. (5) can be given by*

$$o_j = \begin{cases} \sum_{q=1}^{i} \mathbf{A}_{i,q}^{L} \mathbf{V}_{q,j}, & \text{if } i \leq g; \\ \sum_{p=1}^{m} \mathbf{A}_{i,p}^{G} \sum_{q=1}^{g} \Phi_{p,q} \mathbf{V}_{g(p-1)+q,j} + \sum_{t=1}^{w} \mathbf{A}_{i,t}^{L} \mathbf{V}_{mg+t,j}, \ w=s+(i-s) \bmod g, & \text{else,} \end{cases} \quad (6)$$

*where $o_j$ is the $j$-th element of the output $\mathbf{o}$, $\Phi \in \mathbb{R}^{m \times g}$ is the weight of all core tokens in Eqn. (2), $\mathbf{A}$ denote the attention score of CCA-Attention in Eqn. (5).*

*Proof.* We decompose the $i$-th row of the attention scores $\mathbf{A}$ into two parts:

$$\mathbf{A}_i = \left[\mathbf{A}_i^G, \mathbf{A}_i^L\right], \quad (7)$$

where $\mathbf{A}_i^G \in \mathbb{R}^m$ and $\mathbf{A}_i^L \in \mathbb{R}^w$ with $w=s+(i-s) \bmod g$. We aim to use these two terms to formalize $\mathbf{A}_i$ element by element into the structure of full attention. For simplicity, we expand each element in these two terms with the weight $\Phi$ although the $\Phi$ is not directly weighted on the attention:

$$\mathbf{A}_i = \left(\underbrace{\Phi_{1,1}\mathbf{A}_{i,1}^G, \ldots, \Phi_{1,g}\mathbf{A}_{i,1}^G}_{g}, \ldots, \underbrace{\Phi_{m,1}\mathbf{A}_{i,m}^G, \ldots, \Phi_{m,g}\mathbf{A}_{i,m}^G}_{g}, \underbrace{\mathbf{A}_{i,1}^L, \ldots, \mathbf{A}_{i,w}^L}_{w}\right), i=1,\ldots,L. \quad (8)$$

Based on the property of the attention scores of $\mathbf{A}^G$, for $\forall i > j$, we have $a_{ij} > 0$, satisfying the condition of *reachability*. Next, we drive the output representation $\mathbf{o}$ of a token.

When $i \leq g$, for each $o_j$ in $\mathbf{o}$, we can use both the attention score of the locality-preserving module and the globality-aware pooling module to obtain

$$o_j = \sum_{q=1}^{i} \mathbf{A}_{i,q}^{L} \mathbf{V}_{q,j}. \quad (9)$$

When $i > g$, for each $o_j$ in $\mathbf{o}$, we can use the attention score of the locality-preserving module to obtain

$$o_j = \sum_{p=1}^{m} \mathbf{A}_{i,p}^{G} \sum_{q=1}^{g} \Phi_{p,q} \mathbf{V}_{g(p-1)+q,j} + \sum_{t=1}^{w} \mathbf{A}_{i,t}^{L} \mathbf{V}_{mg+t,j} \quad (10)$$

Taking Eqn. (8) and Eqn. (10) together, we obtain the results. $\square$

# B. More Implementation Details

## B.1. More Details on Dataset and Evaluation Metrics

**SlimPajama (Cerebras, 2024)** dataset is an open-source reproduction of the data mixture used to pretrain the LLaMA models. It consists of 82% web data, 4.5% code from Github, 4.5% Wikipedia, 4.5% books, 2.5% Arxiv, and 2.0% StackExchange, used for extending the context lengths of LLMs to 128K tokens through careful data engineering techniques like per-source length upsampling. We use the SlimPajama dataset (Cerebras, 2024) as our training dataset.

**LongBench (Bai et al., 2023)** is a pioneering benchmark designed for the bilingual, multitask, and comprehensive assessment of the long context understanding capabilities within LLMs. It encompasses diverse languages, specifically Chinese and English, thereby facilitating a more exhaustive evaluation of the multilingual proficiencies of large models in long context scenarios. Moreover, LongBench is structured with 6 major categories and 21 distinct tasks, spanning crucial long-text application areas such as single-document QA, multi-document QA, summarization, few-shot learning, synthetic tasks, and code completion. LongBench has 14 English tasks, 5 Chinese tasks, and 2 code tasks. The average length of the majority of tasks falls within the range of 5k to 15k, and it comprises a total of 4,750 test data. For detailed statistical information and construction methodologies of LongBench tasks, reference can be made to the designated source. Additionally, LongBench-E is a test set featuring a more evenly distributed length constructed through uniform sampling. It contains comparable data quantities in the 0-4K, 4K-8K, and 8K+ length intervals, enabling an in-depth analysis of the model's performance fluctuations across different input lengths. We conduct the experiments on LongBench-E to verify the long context understanding capability of models in Section 4.2.

**Exact Match Score (EM Score) (Liu et al., 2024b)** measures the model's ability to find the key information within a long context in a multi-document question-answering task. In this task, each test sample comprises a certain number of documents to reach the specified context length (20 for 4K, 48 for 8K, 96 for 16K, 190 for 32K, 378 for 64K, 755 for 128K), followed by a question. We evaluate EM score metric with the multi-document question-answering dataset in Lost in the Middle (Liu et al., 2024b), which is collected from NaturalQuestions-Open and Wikipedia. We use the exact match score as the evaluation metric, judging whether any of the correct answers appear in the predicted output in Section 4.2.

**Massive Multitask Language Understanding (MMLU) (Hendrycks et al., 2021)** dataset is designed to assess the capabilities of language models across a wide array of subjects, delving deeper into their academic and professional understanding. The MMLU benchmark spans 57 diverse subjects, ranging from elementary mathematics to professional law. The questions are designed to test both world knowledge and problem-solving abilities, challenging models with content from elementary to advanced professional levels. We use the MMLU metric to evaluate the model's proficiency across a diverse set of language-understanding tasks. It tests the model's ability to apply its knowledge to a broad spectrum of topics and question types, reflecting its generalization capability in real-world scenarios. The MMLU metric (Hendrycks et al., 2021), which tests world knowledge and problem-solving abilities in zero-shot and few-shot settings, is evaluated using the MMLU dataset (Hendrycks et al., 2021). This dataset spans 57 subjects across disciplines such as STEM, humanities, and social sciences. We test the MMLU metric in a 5-shot setting with MMLU dataset (Hendrycks et al., 2021) to verify the commonsense generalization ability of models in the supplementary materials C.3.

**Perplexity (PPL)** quantifies how effectively a model can predict the context. It is calculated as the exponentiated average negative log-likelihood of a sequence, offering a statistical measure of language modeling performance. **Proof-pile (Azerbayev et al., 2022)** is a 13GB high-quality dataset of mathematical text and code that comprises 8.3 billion tokens (measured by the gpt-neox tokenizer). The dataset is composed of diverse sources of both informal and formal mathematics and the raw data are downloaded from the web. We report PPL on the test dataset. We use the test dataset of Proof-pile to verify long-context language modeling ability of models in the supplementary materials C.5.

## B.2. More Experimental Protocols

**CCA-Attention (Ours).** For the continuous pretraining, we adopt the SlimPajama (Cerebras, 2024) dataset, an open-source replication of the LLaMA pretraining data mixture. This dataset comprises 82% web data, split between 67% from CommonCrawl and 15% from C4, alongside 4.5% from GitHub code, 4.5% from Wikipedia, 4.5% from books, 2.5% from Arxiv, and 2.0% from Stack Exchange. We replace the full self-attention in LLaMA2 with our proposed CCA-Attention. The number of groups in globality-aware attention is shared across different model sizes. Training is conducted on 8 × A800 GPUs using a micro-batch size of 1 and a gradient accumulation of 8, with a total of 1000 training steps. This training

configuration is applicable to all model sizes and context lengths. Our method requires finetuning on a modest number of tokens to extend the long-context capabilities of LLMs, enabling efficient attention computation. Specifically, we require only 2.10 billion tokens for 32K and 5 billion tokens for 80K, which is significantly lower than the token requirements for retraining a large language model.

To scale the models to long contexts, we modified the "base frequency" in RoPE from 10,000 to 500,000, following (Cerebras, 2024; Xiong et al., 2024). In the globality-aware attention, we set the position embedding of $\mathbf{K}^{global}$ to the position embedding of the token at the middle position in the corresponding group, ensuring that our attention maintains positional awareness. Following FlashAttention (Dao, 2024), we implement our CCA-Attention by leveraging Triton (Tillet et al., 2019) to perform low-level operator fusion between our globality-aware pooling and locality-preserving modules. This enables us to integrate our CCA-Attention as a standalone, cache-friendly operator, effectively eliminating redundant computations.

**Compared Methods**. We conduct comprehensive comparisons between our proposed CCA-Attention and several state-of-the-art methods, including LLaMA-2 with vanilla attention, StreamingLLM (Xiao et al., 2024b), LM-infinite (Han et al., 2023), and MInference (Jiang et al., 2024), across the LongBench (Bai et al., 2023) and RULER (Hsieh et al., 2024) benchmarks. Our experiments are based on the LLaMA-2 7B model fine-tuned on sequences of length 32K and 80K (Fu et al., 2024).

For StreamingLLM (Xiao et al., 2024b), we use the official implementation, adjusting the attention sink to 4 and setting the attention context size to 2000. Similarly, for LM-infinite (Han et al., 2023), we follow the official code, configuring the local branch size to 1024 and the global branch size to 16. In the case of MInference (Jiang et al., 2024), we also employ the official code implementations, configured with the official settings.

# C. More Experimental Results

## C.1. Experiments on More Long Context Benchmarks

We further evaluated the long-context modeling performance of our proposed method on RULER (Hsieh et al., 2024) using the LLaMA2-7B-80K model across various context lengths (ranging from 8K to 64K). As shown in Table 3, our approach consistently outperforms MInference (Jiang et al., 2024) at all context lengths, demonstrating its superiority in context modeling. The performance gains primarily stem from two key innovations in our method: 1) Globality-aware Pooling Module dynamiclly identifies and pools the task-relevant context into core tokens, which effectively reduces redundancy while preserving essential information; 2) A locality-preserving module that supplements local information and ensures comprehensive information interaction across all context segments, as opposed to simply discarding tokens.

Table 4. Comparisons on RULER across 8-64K context.

| Methods | 8K | 16K | 32K | 64K | Avg.↑ |
|---|---|---|---|---|---|
| LLaMA2-7B-80K (*Vanilla Self-Attention*) | **71.90** | 66.26 | **61.54** | **55.15** | **63.71** |
| MInference (Jiang et al., 2024) | 67.78 | 65.32 | 61.43 | 52.77 | 61.83 |
| CCA-LLM (Ours) | 68.15 | **66.31** | 60.89 | 54.88 | 62.56 |

## C.2. Comparisons with More Efficient Attention Methods

To further evaluate our method, we compare it with LongLoRA (Chen et al., 2024), a recently proposed training-based approach with PI techniques. As shown in Table 5, on LongBench-E, our CCA-LLM achieves superior performance in terms of modeling accuracy, inference speed, and memory efficiency. Notably, while S2-Attention proposed by LongLoRA is only available during training, it defaults to full self-attention during inference, resulting in comparable computational and memory overhead to standard attention mechanisms. In contrast, CCA-LLM achieves a first-token latency reduction of 3.5× and a 46% decrease in memory usage, demonstrating its effectiveness for efficient long-context modeling.

Table 5. Comparisons on LongBench-E. "FTL" denotes the latency to generate the first token.

| Model | Avg. Score ↑ | FTL ↓ (s) | Memory. ↓ (GB) |
|---|---|---|---|
| LLaMA2-7B-32K | 22.11 | 9.15 | 35.58 |
| • LongLoRA | 21.58 | 9.15 | 35.58 (0%↓) |
| • CCA-LLM (Ours) | 21.86 | 2.59 | 19.12 (46%↓) |

## C.3. More Comparisons on MMLU with Multi-choice QA

When applied to tasks with a fixed input length, such as multi-choice QA tasks, we set the number of groups $m$ as a constant value. This ensures that the overall computational complexity of our method is $O(L)$, where $L$ represents the input sequence length. In this section, we compare the performance of our method with full self-attention on the MMLU dataset specifically for multi-choice QA tasks. As shown in Table 6, our method consistently achieves superior performance than the traditional self-attention method, demonstrating the effectiveness of our approach.

Table 6. Comparisons on MMLU with multi-choice QA.

| Method | LlaMA2-7B-8K | | LlaMA2-7B-16K | | LlaMA2-13B-16K | | LlaMA2-13B-32K | |
|---|---|---|---|---|---|---|---|---|
| | Full Self-attention | Ours | Full Self-attention | Ours | Full Self-attention | Ours | Full Self-attention | Ours |
| MMLU | 33.34 | **37.55** | 28.19 | **39.71** | 27.17 | **48.11** | 26.72 | **47.93** |

## C.4. Statistical Results of Sparse Attention Scores

We visualized LLaMA2-7B's attention scores on a sentence of 32 tokens in Figure 5 as a supplement to Figure 1. As shown in the figure, these attention scores show consistent sparsity from shallow to deep layers. Same as demonstrated in existing methods (Beltagy et al., 2020; Xiao et al., 2024b).

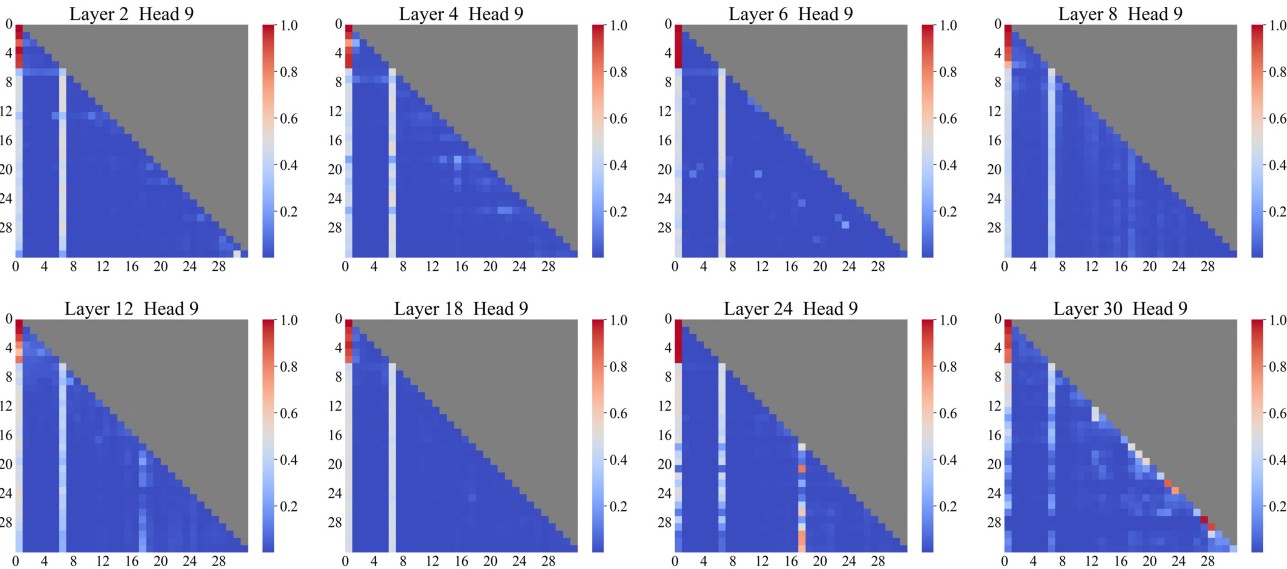

*Figure 5.* A visualization of attention scores in LLaMA2-7B with a sentence of 32 input tokens. The attention map reveals a distinct pattern: the majority of tokens exhibit minimal attention scores. Conversely, a minority of tokens are associated with significantly higher attention scores. This trend is observed consistently from the shallow to the deeper layers of the model.

Based on these observations, our CCA-Attention of assessing token importance within each group using the attention from the last token is both rational and effective. The attention map visualization reveals a distinct pattern where tokens that are important to the query receive consistently high attention scores from all subsequent tokens. This indicates that important tokens, regardless of their position within a group, have a notable influence on attention distribution, suggesting that our method of importance assessment is capable of capturing these crucial tokens.

### C.5. More Ablation Studies

**Effect of Group-wise Pooling Strategy**. For computational efficiency, we conduct ablations with LLaMA2-7B-16K. We adopt perplexity (PPL) to evaluate our CCA-Attention models. To investigate the effect of different pooling strategies, we conduct ablations with max pooling, mean pooling and our weighted pooing in Eqn. (2). In Table 7, our CCA-Attention with group-wise attention pooling strategy achieves superior results in both PPL (*e.g.*, 2.85 *vs.* 2.99). This advantage arises since max pooling retains only the token with the highest response, thereby discarding the semantic importance of the remaining tokens. Mean pooling averages all tokens within a group, which substantially dilutes the semantic significance of critical tokens. In contrast, our CCA-Attention dynamically assigns aggregation weights of each token, facilitating a more comprehensive and efficient fusion.

*Table 7.* Effect of pooling strategy.

| Strategy | Mean Pooling | Max Pooling | CCA-Attention (Ours) |
|---|---|---|---|
| PPL $\downarrow$ | 2.99 | 2.99 | 2.85 |

**Effect of Group Size** $g$. To investigate the effect of different group sizes $g$, we implement the proposed CCA-Attention with different $g \in \{2, 4, 8, 16, 32, 64\}$. In Table 8, as $g$ increases, the computational efficiency improves while the PPL increases. Upon closer examination, the smallest group size $g$ captures the most comprehensive information, which translates to the highest computational cost but also the optimal PPL. Conversely, an excessively large $g$ leads to an overemphasis on globality-aware attention, compressing information to the point where crucial semantic nuances may be overlooked, thereby curtailing performance. To strike a balance between computational efficiency and model performance, we have selected $g{=}16$ as the default training setting.

**Effect of Local Window Size** $s$. To systematically evaluate the influence of different local window sizes $w$, we implement the proposed CCA-Attention across a range of $s \in \{256, 512, 1024, 2048, 4096\}$. In Table 9, an increase in $s$ correlates with

*Table 8.* Effect of group size $g$.

| $g$ | 2 | 4 | 8 | 16 | 32 | 64 |
|---|---|---|---|---|---|---|
| PPL ↓ | 2.75 | 2.78 | 2.81 | 2.86 | 2.90 | 2.93 |
| Latency ↓ (ms) | 546.7 | 503.5 | 479.6 | 462.8 | 457.74 | 456.0 |

lower PPL, but this is counterbalanced by a rise in computational cost. A larger $s$ captures more contextual information with neighborhood tokens, but also increases computational demands. Conversely, a smaller $s$, indicative of a limited receptive field, constrains the exchange of information within the locality-preserving module, resulting in diminished performance. Striking a balance between computational efficiency and model efficacy, we opted for $s=1024$ as the default training setting.

*Table 9.* Effect of local window size $s$.

| $s$ | 256 | 512 | 1024 | 2048 | 4096 |
|---|---|---|---|---|---|
| PPL ↓ | 2.98 | 2.92 | 2.86 | 2.79 | 2.73 |
| Latency ↓ (ms) | 457.4 | 460.1 | 461.4 | 462.8 | 473.1 |

**Effect of Different Updating Strategies**. As mentioned in Section 3.4, we have two updating strategies: 1) updating all the parameters during finetuning (full finetuning) and 2) only updating the parameters $\mathbf{W}^Q$, $\mathbf{W}^K$, $\mathbf{W}^V$ (partial finetuning). In Table 10, we compare these two variants of our methods on the Longbench-E benchmark. The variant that updates all parameters during finetuning achieves better performance because it allows the model to fully adapt to our proposed attention. In contrast, partial fine-tuning, while limiting the model's adaptability due to fixed pre-trained features, still achieves competitive performance. This makes partial fine-tuning a practical choice in scenarios requiring rapid training or where computational resources are limited. Despite its constraints, partial fine-tuning can deliver performance close to that of full fine-tuning, offering a balanced trade-off between efficiency and accuracy.

*Table 10.* Effect of different updating strategies.

| Strategies | Single-Doc. QA | Multi-Doc. QA | Sum. | FS. Learning | Synthetic | Code | Avg. |
|---|---|---|---|---|---|---|---|
| Partial Finetuning | 5.39 | 3.62 | 9.21 | 60.41 | 1.34 | 51.77 | 21.96 |
| Full Finetuning | 5.62 | 4.34 | 8.99 | 59.60 | 0.48 | 54.40 | **22.24** |

## C.6. Training Convergence Curve

In the experiments, we finetune the LLaMA2-7B-32K and LLaMA2-7B-80K models equipped with our CCA-Attention on SlimPajama (Cerebras, 2024) dataset for 1,000 iterations. We show the training convergence curves of both models with our CCA-Attention in Figure 6. From the results, by minimizing the training loss, both LLaMA2-7B-32K and LLaMA2-7B-80K models are able to converge very fast. The perplexity rapidly converges within approximately the first 100 iterations and remains stable over 1,000 iterations. These results not only demonstrate the effectiveness and training stability of our proposed CCA-Attention, but also establish it has the potential to be a plug-and-play attention module incorporated into existing LLMs. Notably, the initial training loss of LLaMA2-7B-32K is higher than that of LLaMA2-7B-80K. This difference arises because LLaMA2-7B-32K is finetuned from the official LLaMA2-7B-4K model, which has a shorter context window and thus requires more significant adjustments to adapt to longer sequences. In contrast, LLaMA2-7B-80K is fine-tuned from a model pre-trained by Fu. et.al. (Fu et al., 2024).

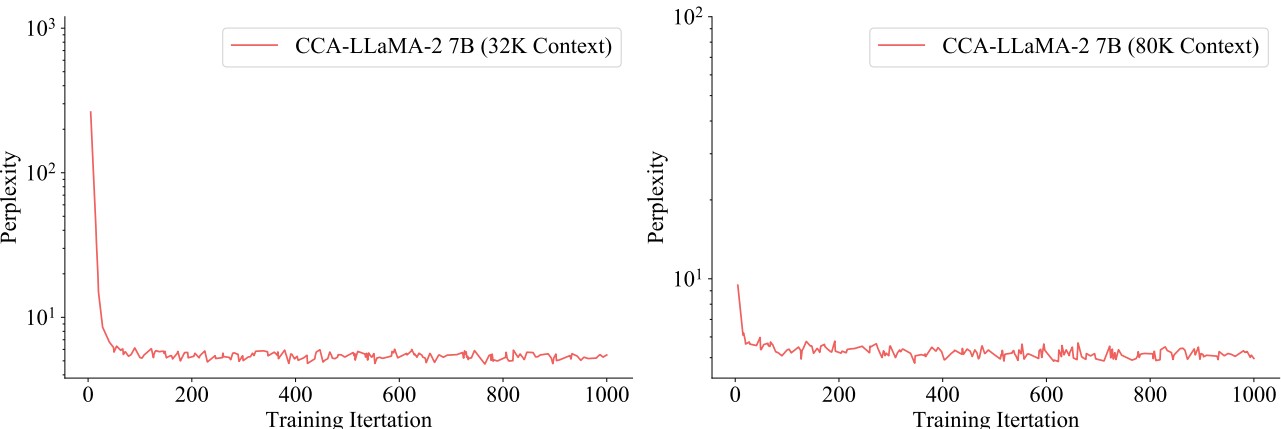

*Figure 6.* Convergence curves of our CCA-LLM models under different contexts.

