# OpenReview forum: "Core Context Aware Transformers for Long Context Language Modeling"
_ICML.cc/2025/Conference — ICML 2025 poster_

### Official Review · Reviewer_XMmi · 2025-03-04

**Overall Recommendation:** 3

**Summary:**

This paper addresses the challenge of long-context language modeling by proposing a Core Context Aware Attention (CCA-Attention) mechanism. The method improves efficiency by dynamically selecting core tokens within token groups while preserving local information. CCA-Attention consists of two main components: (1) a Globality-aware Pooling Module, which segments the input into groups and selects representative core tokens to approximate global attention, and (2) a Locality-preserving Module, which maintains a local attention window to retain fine-grained token interactions. Experimental results demonstrate improvements in both latency and memory usage.

**Claims And Evidence:**

While efficiency gains are demonstrated, the actual performance improvement in terms of model accuracy/quality is "marginal" compared to baselines. Without seeing the exact numbers, it appears the efficiency-performance tradeoff may not be as compelling as suggested.
Generalizability claims: The experiments are limited to two LLaMA-7B models, making it difficult to conclude that these benefits would extend to other architectures or larger models.
Methodological effectiveness: Several technical aspects lack sufficient justification:
The rationale for fixed-size grouping in the pooling module
The potential information loss when using a single core token to represent groups with multiple high-attention tokens
The ability to perform well across varying parameter settings when trained on fixed values
Benchmark performance: The reviewer highlights significant discrepancies between reported results and official leaderboard scores on LongBench tasks, calling into question the validity of the performance evaluation.
Technical clarity: Several implementation details are insufficiently explained, including training strategies, parameter configurations, and variable definitions.

**Essential References Not Discussed:**

The paper only compares against training-free methods, implying that important training-based approaches are missing. For example:
Position Interpolation methods (PI, NTK-aware PI)
LongLLaMA and other specifically fine-tuned long-context models
The absence of these references, particularly other competitive training-based methods, creates a significant gap in contextualizing the paper's contributions. By only comparing against a subset of training-free methods (and not the strongest ones), the paper fails to properly position its approach within the current state of the art for long-context modeling, potentially overstating its relative advantages.

**Experimental Designs Or Analyses:**

The experimental design appears to have several validity concerns, particularly regarding:
- Transparency of experimental parameters
- Benchmark result discrepancies
- Limited model testing
- Unclear training methodologies
- Potentially biased baseline comparisons

These issues substantially impact the ability to verify the paper's claims.

**Methods And Evaluation Criteria:**

The methods and general evaluation approach make sense for the problem of efficient long-context modeling, but several gaps in the evaluation methodology and presentation limit the strength of conclusions that can be drawn from the results. The paper would benefit from more comprehensive model testing, clearer parameter specifications, and better explanation of observed performance patterns.

**Other Comments Or Suggestions:**

1. Some variable definitions are unclear. For example, in line 225, what does s represent? Although Algorithm 1 mentions it as the local window size, it should be explicitly defined the first time it appears in the main text. Similarly, how is K_1^G obtained? Does it refer to the first dimension of K^G? A clearer explanation is needed.

**Other Strengths And Weaknesses:**

Strengths
1. The paper focuses on an important and timely research problem—efficient long-context modeling.
2. Combining the long-range Globality-aware pooling module with the short-range Locality-preserving module intuitively makes sense and is quite innovative.
3. The paper is also well-written, and the illustrations are clear and detailed, making it easy to follow.
﻿
Weaknesses
1. The baselines used for comparison are all training-free methods, yet the average performance improvement is relatively small. While the proposed method achieves better latency and memory efficiency, the actual improvement in average score is marginal. A discussion on why this is the case would be helpful.
2. Experiments are only conducted on two LLaMA-7B models. To better demonstrate the generalization of the proposed method, it would be beneficial to evaluate it on larger models or different architectures.
3. What are the benefits of using fixed-size grouping in the Globality-aware pooling module? If multiple tokens within a group have high attention scores with the last token, wouldn't using one core token to represent them introduce significant approximation errors?
4. The scores in Table 1 seem unreasonable for tasks such as single document QA, multi-document QA, and summarization because there is a significant discrepancy compared to the official leaderboard in https://github.com/THUDM/LongBench/tree/main/LongBench. Could the authors clarify the reason for this difference?
5. In Figure 3, CCA-LLM uses different group size g and local window size s. The authors state that the group size g ranges from 16 to 2, and the local window size s ranges from 4096 to 1024. However, the specific parameter values used for each point are not annotated in the figure, which causes confusion given the apparent significant differences in performance. Additionally, I'm curious whether during the fine-tuning only fixed g and s are used; how can it perform well across varying g and s settings?
7. CCA-LLM’s training strategy in Table 1 and Table 2 is unclear. Was full fine-tuning or partial fine-tuning used? It would be helpful if the authors could provide results for different fine-tuning strategies to allow readers to better understand the trade-off between training cost and model performance.

**Questions For Authors:**

1. What is the batch size in the experiment shown in Figure 4? It appears that CCA-LLM can handle context lengths up to 128K without significantly increasing kv cache usage compared to full self-attention. What are the configurations of s and g here? Moreover, MInference barely reduces kv cache usage, which is also confusing. Could the authors clarify it?

**Relation To Broader Scientific Literature:**

The paper make a relevant contribution to the efficient long-context modeling literature, though with the methodological and evaluation limitations.

**Theoretical Claims:**

The paper takes a primarily experimental approach to validating its method, with theoretical understanding coming from empirical results rather than formal proofs.

---

> ### Author Rebuttal · Authors · 2025-04-01
>
> > Q1. The paper only compares against training-free methods, important training-based approaches are missing. While the method achieves better latency and memory efficiency, the actual improvement in average score is marginal.
>
> A1. We appreciate your emphasis on contextualizing our work against training-based methods. Below, we address the concerns raised:
>
> **Method Positioning and Performance**
> Our work focuses on **long-context modeling via efficient attention mechanism** for pretrained LLMs. This is distinct from position interpolation methods or long-context adaptation techniques like LongLLaMA, which do not modify attention in LLMs. By reducing computational costs ( **7.9×** on 128K context) while outperforming training-free baselines, our method offers immediate value to practitioners. Last, we emphasize our goal is to reduce redundancy in attention, which should be compared with those that also optimize attention (e.g., MInference).
>
> **Further Experiments**
> To address your concerns, we further compare our method with a training-based model, LongLoRA[ICLR 2024] with PI techniques. In Table I, **our CCA-LLM outperforms LongLoRA in modeling accuracy, inference speed and memory footprint**. We will include these in the revised manuscript.
>
> Table I: Results on LongBench-E. ''FTL'' denotes the latency to generate the first token.
> |Model|Avg.↑|FTL↓ (s)|Mem.↓ (GB)|
> |-|-|-|-|
> |LLaMA2-7B-32K|22.11|9.15|35.58 |
> |• LongLoRA|21.58|9.15 (1.0x)|35.58 (0%↓)|
> |• CCA-LLM (Ours)|**21.86**|**2.59** (**3.5**x)|**19.12** (**46**%↓)|
>
> ** Note: During inference, LongLoRA exhibits the same computational and storage overhead as full self-attention. The reason is LongLoRA's S2-Attention is only applicable in training and uses full self-attention during inference.
>
>
> > Q2. More results on different architectures.
>
> A2. According to your suggestions, we conduct more experiments on LLaMA3.1-8B and Qwen2.5-7B (128K context). The results in Tables II and III show CCA-Attention outperforms state-of-the-art counterpart MInference in terms of **performance, computational efficiency**, and **memory reduction** (see Table II and III).
>
> We would include these results in the revised paper.
>
> Table II. Comparisons of Llama3.1-8B-Instruct-128K on LongBench-E.
> |Model|Avg.↑ (%)|FTL↓ (s)|Mem.↓ (GB)|
> |-|-|-|-|
> |Llama3.1-128K|37.93|9.55|40.38|
> |• MInference|37.74|4.93 (1.9x)|35.95(11%↓)|
> |• **CCA-LLM (Ours)**|**37.81**|**3.08** (**3.1**x)|**20.63** (**49**%↓)|
>
> Table III. Comparisons of Qwen2.5-7B-128K on LongBench-E.
> |Model|Avg.↑ (%)|FTL↓ (s)|Mem.↓ (GB)|
> |-|-|-|-|
> |Qwen2.5-128K|38.38|10.58|35.11|
> |• MInference|36.72|4.86 (2.2x)|32.40 (8%↓)|
> |• **CCA-LLM (Ours)**|**38.08**|**2.74** (**3.9x**)|**19.31** (**45**%↓)|
>
> > Q3. Benefits of using fixed-size grouping.
>
> A3. This allows for optimized memory access and parallel computation. While dynamic group sizes would introduce significant implementation complexities.
>
> > Q4. If multiple tokens in a group have high scores with the last token, wouldn't using one core token to represent them introduce significant approximation errors?
>
> A4. The mentioned issue can be significantly mitigated by the fine-tuning process, based on the finding: CCA attention with/without fine-tuning yields performance of 6.92 and 22.24 on LongBench-E, respectively. This confirms core token aggregation effectively captures group-wise information.
>
>
> > Q5. Why are the scores in Table 1 different from the official leaderboard?
>
> A5. The leaderboard tests **chat models** with SFT on LongBench, while we test **base models** without SFT on LongBench-E.
>
> > Q6. Whether during the fine-tuning only fixed g and s are used? How can it perform well across varying g and s?
>
> A6. We use fixed $g$ and $s$ during finetuning. The reasons for the promising performance with varying $g$ and $s$ are twofold:
> - Globality-aware pooling module dynamically **pools groups of tokens into core tokens based on their importance, regardless of $g$**. Thus it generalizes to arbitrary $g$ during inference.
>  - Locality-preserving module employs rotary position embeddings that encode relative distances rather than absolute positions, which **decouples from specific $s$**, allowing adaptation to different $s$.
>
> We would include these in the revised paper.
>
> > Q7. Was CCA-LLM fully or partially fine-tuned in Tables 1 & 2? Could you compare them?
>
> A7. Full fine-tuning. In Section C.3, we compare different fine-tuning strategies. Full fine-tuning achieves better performance than partial fine-tuning (22.24 vs. 21.96) on LongBench-E with more training hours (15.4 vs. 11.2).
>
> > Q8. Implementation details in Figure 4.
>
> A8. We use batch size=1, $g$=16 and $s$=1024.
>
>
> > Q9. Why does MInference barely reduce KV cache usage?
>
> A9. Minference's sparse attention is only applicable in pre-filling. During decoding, it uses full self-attention, leading to same KV cache usage.
>
> ---
> We sincerely hope our clarifications above have addressed your questions.

---

### Official Review · Reviewer_JpcZ · 2025-03-09

**Overall Recommendation:** 2

**Summary:**

This work proposes Core-Context-Aware attention to enhance efficient long-context modeling for LMs. Two modules are included: a global pooling module to compresses groups of tokens into core tokens and a local module to preserve local information. Experiments on various tasks and datasets show the effectiveness of the proposed approaches.

Update after rebuttal:

I do appreciate the authors' promise of improvements and extra results. Nevertheless, I think my major concern over the similarities between this work and previous work from the perspective of methodology is still not addressed, which is why I will keep my original score.

**Claims And Evidence:**

Mostly.

**Essential References Not Discussed:**

There have been a line of context-compression methods not fully discussed and compared:

Pooling-styled compression:
- Compressive Transformer: https://arxiv.org/pdf/1911.05507

Special-token based compression:
- Gist: https://arxiv.org/pdf/2304.08467
- Nugget: https://arxiv.org/pdf/2310.01732
- Landmark: https://arxiv.org/pdf/2305.16300
- Beacon: https://arxiv.org/pdf/2401.03462
- Dodo: https://arxiv.org/pdf/2310.02409

**Experimental Designs Or Analyses:**

The experimental designs seem reasonable to me.

**Methods And Evaluation Criteria:**

Yes.

**Other Comments Or Suggestions:**

- It might be interesting to explore more long-context tasks in addition to LongBench. Recently there have been a variety of long-context benchmarks such as RULER and HELMET, which cover a wider range of tasks and types of contexts.
- Llama 2 is used as the base model, which has a limited context window. It would be interesting to explore whether the proposed approach can help more recent models that have already extended to long contexts (such as Llama 3 and Qwen models).

**Other Strengths And Weaknesses:**

Strengths:

- The paper is overall easy to understand.
- The proposed methods have been shown effective for various tasks and datasets.

Weaknesses:
- The proposed methods share a variety of similarites to previous directions of context compression, some of which are not well discussed and compred with.
- There is a lack of ablation studies and analyses on some design factors of the proposed approach.

**Questions For Authors:**

- In the proposed method, there are some key hyper-parameters that need to be decided, such as the group size and local window size. I'm wondering how they are decided and what are their effects on the model's performance? It would be nice to include more analyses and ablation studies.
- I'm also wondering how much data will be needed to fine-tune the model with the proposed approach?

**Relation To Broader Scientific Literature:**

N/A

**Theoretical Claims:**

N/A

---

> ### Author Rebuttal · Authors · 2025-04-01
>
> > Q1. The proposed methods share similarities to previous context compression methods[r1-r6]. r1: Compressive Transformers, r2: Gist, r3: NUGGET, r4: Landmark, r5: Beacon, r6: DODO.
>
> A1. Thanks for your valuable comments. We clarify the differences below:
>
> - **Problem Importance & Motivation**
> The core challenge we tackle is efficient and effective long-context modeling in LLMs. While self-attention handles long sequences, its quadratic complexity and redundancy in ultra-long contexts (e.g., 128K) severely degrade efficiency and modeling performance. Unlike prior works either compress features via auxiliary networks [r1] or compress context with extra specific tokens [r2-r6]), our method aims to dynamically identify and strengthen task-relevant core-context while suppressing redundancy. This distinction is crucial: compression prioritizes shortening sequences, whereas our core-context awareness optimizes context redundancy in self-attention computation for long context modeling.
> - **Novelty & Differences**
> Our CCA Attention introduces two innovative, complementary mechanisms:
>   - Globality-aware Pooling Module: Unlike context compression methods [r1-r6], it dynamically groups tokens and pools them into core tokens based on their importance. This ensures the model **focuses on task-relevant context while reducing redundancy**.
>   - Locality-preserving Module: While prior compression methods may have a risk of losing local details, our module selectively preserves neighboring tokens to maintain locally relevant information.
>
>
> **We would cite the above papers and include these discussions in the revised manuscript.**
>
> ---
>
> > Q2. It might be interesting to explore more long-context tasks in addition to LongBench, such as RULER.
>
> A2. According to your suggestions, we conduct more experiments on RULER on LLaMA2-7B-80K with varying context lengths (4K-64K). From the results in Table I, our method achieves **superior performance over MInference, a state-of-the-art counterpart**.
>
> Table I: Comparisons of LLaMA2-7B-80K on RULER.
> |Methods|8K|16K|32K|64K|Avg.↑|
> |-|-|-|-|-|-|
> |LLaMA2-80K|71.90|66.26|61.54|55.15|63.71|
> |• MInference|67.78|65.32|**61.43**|52.77|61.83|
> |• CCA-LLM|**68.15**|**66.31**|60.89|**54.88**|**62.56**|
>
> > Q3. It would be interesting to explore whether the proposed approach can help more recent models (such as LLama 3 and Qwen models).
>
> A3. We conduct more experiments on LLaMA3.1-8B and Qwen2.5-7B with context length 128K in Tables II and III, respectively. From the results, our method is able to help LLaMA3.1 and Qwen models improve computational efficiency and memory reduction in long-context modeling.
> - **Computational efficiency**: CCA-Attention achieves 3.1× on LLaMA3.1-8B, and 3.9× on Qwen2.5-7B with 32K context.
> - **Memory reduction**: CCA-Attention reduces 49% GPU memory footprint on LLaMA3.1-8B, and 45% on Qwen2.5-7B.
> - **Performance**: CCA-Attention consistently outperforms MInference across all models, maintaining competitive accuracy with valina self-attention.
>
> Table II. Comparisons of Llama3.1-8B-Instruct-128K on LongBench-E. ''FTL'' denotes the latency to generate the first token in the pre-filling stage.
> |Model| Avg.↑ (%)|FTL↓ (s)|Mem.↓ (GB)|
> |-|-|-|-|
> |Llama3.1-128K|37.93| 9.55 |40.38 |
> |• MInference|37.74|4.93 (1.9x)|35.95(11%↓)|
> |• **CCA-LLM (Ours)**|**37.81**|**3.08** (**3.1**x)|**20.63** (**49**%↓)|
>
> Table III. Comparisons of Qwen2.5-7B-128K on LongBench-E.
> |Model| Avg.↑ (%)|FTL↓ (s)|Mem.↓ (GB)|
> |-|-|-|-|
> |Qwen2.5-128K|38.38|10.58 |35.11 |
> |• MInference|36.72|4.86 (2.2x)|32.40 (8%↓)|
> |• **CCA-LLM (Ours)**| **38.08**|**2.74** (**3.9x**)|**19.31** (**45**%↓)|
>
>
> > Q4. There are some key hyper-parameters, such as the group size and local window size. It would be nice to include more analyses and ablation studies.
>
> A4. In **Section C.3 of supplementary**, we show ablation studies on group size $g$ and local window size $s$. From the results, as $g$ increases, the efficiency will improve and the performance will decrease (in terms of PPL); as $s$ increases, the performance will improve and the efficiency will reduce. To achieve a balance, we select $g=16$ and $s=1024$ as default training setting.
>
> > Q5. How much data will be needed to fine-tune the model?
>
> A5. In **Section B.2 of the supplementary**, we show the details for fine-tuning: We use SlimPajama dataset to fine-tune the model, with only 2.1 billion tokens for LLaMA2-7B-32K and 5.0 billion tokens for LLaMA2-7B-80K.
>
> ---
>
> We sincerely hope our clarifications above have addressed your questions.

---

> > ### Comment · Reviewer_JpcZ · 2025-04-02
> >
> > Thank you for your response.
> >
> > For "Novelty & Differences":
> >
> > > Unlike context compression methods [r1-r6], it dynamically groups tokens and pools them into core tokens based on their importance.
> >
> > I think there are little key differences between many previous work and this work, most of them use a fixed chunking-based sequence segmentation scheme, and reduce the original raw tokens to compressed token representations, which is similar to the method adopted in this work (though the detailed calculation methods may be sligtly different). In the sense of "dynamically selecting", many previous work as discussed before can also achieve this through self-attention.
> >
> > > Locality-preserving Module: While prior compression methods may have a risk of losing local details, our module selectively preserves neighboring tokens to maintain locally relevant information.
> >
> > I think adopting local sliding windows has been a common technique to keep local information, which is hard to be considered as a main novelty point.
> >
> > In this way, I think at least some of these previous work should be considered in the comparisons (at least they should be discussed, which have been ingored in the current version).
> >
> > For the extra experiments on latest llama and qwen models, thanks for the updates and I think these results are important and should be further extended and myabe replace the original results in the current version (also considering LLaMA2-7B-32K and LLaMA2-7B-80K are non-llama-offical extended models).
> >
> > I do appreaciate the authors' extra experiments and also considering other reviewers' judgments, my current evaluation for this work is somewhat borderline (around 2.5, but unfortunately there is no option for this). Please feel free to let me know if there are any points that I may have missed or misunderstood.
> >
> > To make this work more solid (towards a positive score from my side), I think:
> > 1) Comparisons and discussions related to previous work should be carefully included,
> > 2) More extended experiments with the latest models should be adopted as the main results.

---

> > > ### Author Response · Authors · 2025-04-03
> > >
> > > Dear Reviewer,
> > >
> > > Thank you very much for your prompt response to our rebuttal. We hope to make the following further responses to your concerns and sincerely hope you would be satisfied.
> > >
> > > > Q1. Comparisons and discussions related to previous work should be carefully included.
> > >
> > > A1. We further highlight our novelty and clarify the key differences as follows:
> > >
> > > **1. Problem Definition**
> > >
> > > Our work focuses on improving long-context modeling efficiency by reducing redundancy in self-attention. Long-context modeling (e.g., 128K) is important for real-world applications like document analysis, and multi-turn dialogues. However, the quadratic complexity of full self-attention and the redundancy in long contexts pose two key challenges:
> > >
> > >  - **Performance Degradation**: The redundant context may hamper LLMs from capturing dependencies among crucial tokens, degrading representation performance
> > >  - **Unnecessary Computation**: The redundancy in attention introduces unnecessary computational and storage overhead, especially for extremely long contexts.
> > >
> > >
> > > Our proposed **Core Context Aware Attention (CCA-Attention)** addresses redundancy issues in self-attention through:
> > >
> > >  - **Globality-aware pooling module** dynamic pools input tokens into core tokens based on their significance. This enables our method to automatically focus on the core context.
> > >  - **Locality-preserving module** captures the local and fine-grained context by focusing on neighboring tokens. It is not a standalone component but is complementary to the globality-aware pooling module.
> > >
> > > The complementary nature of these modules ensures that **both high-level semantics (via core tokens) and low-level details (via local tokens)** are preserved, while improving the computational and storage effciency.
> > >
> > >
> > > **2. Limitations of Existing Methods**
> > >
> > > Existing approaches [r1-r6] on context compression primarily include two strategies:
> > >
> > >  - Gist[r2], Landmark[r4] and Beacon[r5] compress input/intermediate tokens into smaller sets of special tokens which can be cached and reused for compute efficiency.
> > >  - Compressive Transformers[r1], NUGGET[r3] and DODO[r6] introduce auxiliary networks to compress past tokens into compact representations with reduced dimensions.
> > >
> > > These methods may encounter two key limitations:
> > >
> > >  - They may overlook the sparsity patterns of self-attention in long contexts. In contrast, **our CCA-Attention directly targets redundancy reduction in self-attention**. To this end, we propose a parameter-free group-wise pooling strategy that dynamically pools redundant tokens. Critically, the pooling weights are derived based on context-aware importance, eliminating the need for additional parameters or specialized tokens. **Our method retains full compatibility with existing LLM architectures while improving computational and storage efficiency**.
> > >  - They support acceleration either prefilling [r2,r3,r6] or decoding [r1,r4,r5] phase in inference. In contrast, our proposed CCA-Attention enables efficient long context modeling (e.g., 128K) while **accelerating both training and inference processes (including prefilling and decoding phases)**. CCA-Attention achieves significant speedups (7.9$\times$ speed up over vanilla self-attention on 128K context) while maintaining comparable model performance with a hardware-optimized operator with Triton. Crucially, our method reduces KV cache memory usage by up to 93% in 128K context, improving deployment feasibility without compromising accuracy.
> > >
> > >
> > > **3. Contributions of Our CCA-Attention**
> > >
> > > We would highlight our contributions as follows:
> > >
> > >  - Our CCA-Attention is an **efficient and plug-and-play module for existing LLMs in long-context modeling**. By using core tokens as proxies, CCA reduces attention complexity to linear while requiring minimal fine-tuning for pretrained LLMs.
> > >  - We introduce a dynamic globality-aware pooling module to derive core tokens based on their importance and a local-preserving module to retain local tokens. This combines **both high-level semantics and low-level details, leading to accurate long-term dependency modeling**.
> > >  - Experiments show CCA-Attention outperforms existing efficient attention methods, achieving a 7.9× speedup and a 93% KV cache reduction over full self-attention in 128K context with comparable accuracy.
> > >
> > > We hope this clarification highlights the novelty and contributions of our method. **We will cite these papers and carefully discuss these differences from [r1-r6] in the revised manuscript.**
> > >
> > > > Q2. More extended experiments with the latest models should be adopted as the main results.
> > >
> > > A2. According to your suggestions, we will **update the main results in Tables 1 and 2 to prioritize experiments on LLaMA3.1-8B-128K and Qwen2.5-7B-128K**, as these are officially supported and widely recognized models.
> > >
> > > ---
> > >
> > > We sincerely hope our clarifications above have addressed your concerns. Thank you again for your constructive suggestions to strengthen our paper.
> > >
> > > Best,
> > >
> > > Authors

---

### Official Review · Reviewer_qgiU · 2025-03-14

**Overall Recommendation:** 5

**Summary:**

This paper addresses the computational inefficiency and redundancy in Transformer-based Large Language Models (LLMs) when processing extremely long contexts (e.g., 128K tokens). The authors propose Core Context Aware (CCA) Attention, a plug-and-play mechanism comprising two modules: (1) a globality-aware pooling module that dynamically compresses input tokens into core tokens via weighted grouping, and (2) a locality-preserving module that retains fine-grained local context. The fusion of these modules reduces computational complexity from quadratic to linear while preserving full token reachability. Experiments on LongBench and multi-document QA benchmarks demonstrate superior performance (e.g., 7.9× speedup at 128K context) over methods like MInference and StreamingLLM, with reduced memory footprint.

## update after rebuttal
Thanks to the authors for addressing my concerns and providing additional results.
I didn't find novelty issue on the idea of core context modeling on my side.
I will keep my score.

**Claims And Evidence:**

The claims in the paper are supported by clear evidence.

**Essential References Not Discussed:**

n/a

**Experimental Designs Or Analyses:**

The experimental design is comprehensive, covering a range of benchmarks and metrics to validate the proposed method's effectiveness. The use of LongBench (Bai et al., 2023) and multi-document EM scores (Liu et al., 2024b) provides a robust assessment of long-context understanding. The inclusion of various model sizes (LLaMA2-7B-32K and LLaMA2-7B-80K) ensures scalability analysis. The comparison with state-of-the-art methods (StreamingLLM, LM-Infinite, MInference) highlights the competitive edge of the CCA-Attention mechanism.

I have a few questions regarding the experimental design:
1. The paper explores a fixed group size m for token grouping. While this simplifies experimentation, real-world applications often require balancing accuracy and efficiency across diverse tasks. Without requiring additional experiments, could the authors discuss: is there a principled way to set m for unseen tasks (e.g., smaller groups for high-precision tasks vs. larger groups for efficiency) ?
2. Table 2 shows that model performance does not consistently degrade with increasing text length. A more in-depth analysis or explanation would enhance the understanding of the method.

**Methods And Evaluation Criteria:**

The methods presented in the paper are well-motivated, aiming to balance computational efficiency with model performance. However, I still have some question:

1. Figure 3 demonstrates stable performance with varying group sizes (g) and window lengths (s), what are the possible reasons why changing the size of g and s can work also well during inference?
2. Could the core token (Eqn.2) be biased toward later tokens in a group due to using the last token’s query?

**Other Comments Or Suggestions:**

1. The claim of “significant improvements” seems over-stated. Adding “compared with other baseline methods” would make it more accurate.
2. Is it be clear if $C_t^1(x_t)$ is revised as $C^1(x_t)$?
3. In Figure 2, the bold font usage of g is inconsistent, and $A^G$ is confusing to me.
4. The meaning of “LLaMA2-7B-32K” and “LLaMA2-7B-80K” should be specified.
5. The notations $K$ and $V$ in 12-th in Algorithm 1 should be performed by \tilde.

**Other Strengths And Weaknesses:**

Strengths:

1. The method's primary strength lies in its compatibility with pretrained LLMs, such as LLaMA2. This plug-and-play design minimizes the need for extensive retraining, making it easier to integrate into existing models.
2. CCA integrates Triton, whose ability to optimize low-level operations enhances the method's efficiency, enabling faster inference times.

Weaknesses:
1. The method currently uses a fixed group size for inference, which limits its adaptability to varying text lengths and complexities. The paper lacks an analysis of adaptive grouping strategies, which could improve the method's flexibility and performance.

**Questions For Authors:**

1. will the assumption of redundant text be related to the topic of testing cases? Math and general qa may have different degree of redundancy

**Relation To Broader Scientific Literature:**

Although CCA builds on some sparse attention (eg., Longformer) and linear approximations (eg., RetNet), it introduces dynamic token compression, significantly reducing the computational costs and storage demands for the attention while outperforming other baseline methods like LM-Infinite and MInference.

**Theoretical Claims:**

I have thoroughly reviewed the theoretical claims in the manuscript, which are well-structured and insightful. The authors provide a rigorous reachability guarantee through Proposition 1, demonstrating that the CCA-Attention mechanism preserves full token accessibility by integrating global and local attention scores. This formalization ensures that all tokens remain interconnected, addressing a critical aspect of long-context modeling. Additionally, the computational and storage complexity analysis is comprehensive, highlighting the method's superiority in reducing computation costs and optimizing key-value cache storage. These analyses underscore the method's efficiency and effectiveness for handling long-context tasks.

---

> ### Author Rebuttal · Authors · 2025-04-01
>
> >Q1. Figure 3 demonstrates stable performance with varying group sizes (g) and window lengths (s) during inference, what are the possible reasons?
>
> A1. We deeply appreciate your valuable feedback. We design our CCA-Attention for flexibility, enabling promising performance across varying $g$ and $s$ settings without requiring additional fine-tuning. The reasons are twofold:
>
> - Globality-aware pooling module dynamically **pools groups of tokens into core tokens based on their importance, regardless of the group size $g$**. The pooling weights are derived from intra-group attention scores, which depend on semantic relevance rather than fixed positional patterns. As a result, the module generalizes to arbitrary group sizes $g$ during inference.
>  - Locality-preserving module is inherently translation-invariant to window boundaries by employing rotary position embeddings that encode relative distances rather than absolute positions. This relative distance encoding naturally **decouples the module's operation from specific window size $s$**, allowing adaptation to different $s$ in inference while preserving local context relationships.
>
> We would include these discussions in the revised paper.
>
> > Q2. Could the core token (Eqn.2) be biased toward later tokens in a group due to using the last token’s query?
>
>
> A2. In our CCA-Attention, the strategy of deriving core tokens based on the attention scores from the last token is both rational and effective. This is supported by two points:
>
>  - The attention map visualization in Section C.2 of supplementary material reveals a distinct pattern: **tokens that are important to the query receive consistently high attention scores from all subsequent tokens**. This indicates that important tokens, regardless of their position within a group, have a notable influence on the attention distribution. This suggests that our pooling weights assessed by the last group token are able to capture these crucial tokens.
>  - The consistent high performance in long-context modeling tasks confirms that our CCA-Attention effectively captures global and local dependencies within long contexts. This is a direct result of our pooling strategy, which shows that information relevant to the query is not overlooked.
>
>
> We would include these discussions in the revised paper.
>
> > Q3. Without requiring additional experiments, could the authors discuss: is there a principled way to set m for unseen tasks?
>
> A3. We appreciate the reviewer's insightful comments. In practice, we are able to **set different group sizes according to the nature of the task**. For high-precision tasks (e.g., legal document analysis), smaller group sizes are preferable to ensure finer granularity and capture more detailed contextual dependencies. Conversely, for tasks prioritizing efficiency over precision (e.g., real-time dialogue systems), larger group sizes are able to reduce computational overhead. This trade-off aligns with the inherent flexibility of our proposed CCA-Attention, which allows dynamic adjustment of m during inference.
>
>
> > Q4. Could the authors explain why the model performance does not consistently degrade with increasing text length in Table 2?
>
> A4. As discussed in the paper, we discovered that full self-attention may face redundant context issues in sequence modeling. **This redundancy may hamper the modeling performance** by 1) weakening the focus on semantically important content, and 2) introducing noise from irrelevant context. More importantly, **this issue becomes more severe with longer sequences**.
>
> To address this, we propose a core context aware attention mechanism, in which **non-core/irrelevant contexts will be compressed by weighted pooling**. In this way, CCA-Attention not only alleviates the redundant context issue but thus improves the long-context modeling performance. As a result, the advantages of our method become more prominent as the length of the context increases.
>
> > Q5. Will the assumption of redundant text be related to the topic of testing cases? Math and general QA may have different degrees of redundancy.
>
> A5. **Our assumption of redundant context is not related to the topic**. To verify this, we perform an analysis using LLaMA2-7B on 16K length corpora from two domains: math (MathPile dataset[1]) and general QA (WikiText dataset[2]).
>
> By examining the attention weights of the final token relative to preceding tokens, we quantify the redundancy degree as the proportion of tokens receiving attention weights below 0.001. **The results demonstrate comparable levels of redundancy between math** (99.8% tokens' weights below 0.001) **and general QA context** (98.7% tokens' weights below 0.001), supporting our initial assumption.
>
> [r1] A Billion-Token-Scale Pretraining Corpus for Math. NeurIPS 2024.
>
> [r2] Pointer Sentinel Mixture Models. ICLR 2017.
>
> ---
> We sincerely hope our clarifications above have addressed your concerns.

---

### Decision · Program_Chairs · 2025-05-01

**Decision:**

Accept (poster)

**Comment:**

The paper introduces an innovative approach to long-context language modeling, called Core Context Aware (CCA) Attention, which significantly reduces computational complexity and storage overhead while maintaining full token accessibility. The CCA-Attention mechanism dynamically groups tokens and compresses them into core tokens based on their significance, helping to focus on semantically important content and reducing redundancy. This novel approach is designed to be a plug-and-play mechanism for existing LLMs, offering compatibility and flexibility while achieving superior performance in both computational efficiency and memory usage reduction. Experimental results demonstrate the method's effectiveness, with a 7.9× speedup and a 93% reduction in memory footprint over full self-attention on 128K context, without compromising accuracy.
The main concern about the paper is the novelty which has been discussed in the rebuttal, mainly with reviewer JpcZ.  Although I agree that the paper's novelty is not such significant, I think the plug-and-play feature of the method is new and demonstrates its efficiency compared with existing work.  So I would give a "weak accept" to this paper.